# DISTRIBUTIONAL REINFORCEMENT LEARNING VIA SINKHORN ITERATIONS

## ABSTRACT

Distributional reinforcement learning (RL) is a class of state-of-the-art algorithms that estimate the entire distribution of the total return rather than only its expectation. The empirical success of distributional RL is determined by the representation of return distributions and the choice of distribution divergence. In this paper, we propose a new class of *Sinkhorn distributional RL (SinkhornDRL)* algorithm that learns a finite set of statistics, i.e., deterministic samples, from each return distribution and then uses Sinkhorn iterations to evaluate the Sinkhorn distance between the current and target Bellmen distributions. Sinkhorn divergence features as the interpolation between the Wasserstein distance and Maximum Mean Discrepancy (MMD). SinkhornDRL finds a sweet spot by taking advantage of the geometry of optimal transport-based distance and the unbiased gradient estimate property of MMD. Finally, compared to state-of-the-art algorithms, Sinkhorn-DRL's competitive performance is demonstrated on the suite of 55 Atari games.

## 1 INTRODUCTION

Classical reinforcement learning (RL) algorithms are normally based on the expectation of discounted cumulative rewards that an agent observes while interacting with the environment. Recently, a new class of RL algorithms called *distributional RL* estimates the full distribution of total returns and has exhibited the state-of-the-art performance in a wide range of environments (Bellemare et al., 2017a; Dabney et al., 2018b;a; Yang et al., 2019; Zhou et al., 2020; Nguyen et al., 2020).

In distributional RL literature, it is easily recognized that algorithms based on either Wasserstein distance or MMD have gained great attention due to their superior performance. Their mutual connection from the perspective of mathematical properties intrigues us to explore further in order to design new algorithms. Particularly, Wasserstein distance, long known to be a powerful tool to compare probability distributions with non-overlapping supports, has recently emerged as an appealing contender in various machine learning applications. It is known that Wasserstein distance was long disregarded because of its computational burden in its original form to solve an expensive network flow problem. However, recent works (Sinkhorn, 1967; Genevay et al., 2018) have shown that this cost can be largely mitigated by settling for cheaper approximations through strongly convex regularizers. The benefit of this regularization has opened the path to wider applications of the Wasserstein distance in relevant learning problems, including the design of distributional RL algorithms.

The Sinkhorn divergence (Sinkhorn, 1967) introduces the entropic regularization on the Wasserstein distance, allowing it tractable for the evaluation especially in high-dimensions. It has been successfully applied in numerous crucial machine learning developments, including the Sinkhorn-GAN (Genevay et al., 2018) and Sinkhorn-based adversarial training (Wong et al., 2019). More importantly, it has been shown that Sinkhorn divergence interpolates Wasserstein ditance and MMD, and their equivalence form can be well established in the limit cases (Feydy et al., 2019; Ramdas et al., 2017; Nguyen et al., 2020). However, a Sinkhorn-based distributional RL algorithm has not yet been formally proposed and its connection with algorithms based on Wasserstein distance and MMD is also less studied. Therefore, a natural question is *can we design a new class of distributional RL algorithms via Sinkhorn divergence, thus bridging the gap between existing two main branches of distributional RL algorithms?* Moreover, the dominant quantile regression-based algorithms, e.g., QR-DQN (Dabney et al., 2018b), aimed at approximating Wasserstein distance, suffers from

the non-crossing issue in the quantile estimation (Zhou et al., 2020), while sample-based Sinkhorn distributional algorithm can naturally circumvent this problem.

In this paper, we propose a novel distributional RL family based on *Sinkhorn divergence*. Firstly, we show key roles of distribution divergence and value distribution representation in the design of distributional RL algorithms. After a detailed introduction of our proposed SinkhornDRL algorithm, with a non-trivial proof, we theoretically analyze the convergence property of distributional Bellman operators under Sinkhorn divergence. A regularized MMD equivalence with Sinkhorn divergence is also established, interpreting its empirical success in real applications. Finally, we compare the performance of our SinkhornDRL algorithm with typical baselines on 55 Atari games, verifying the competitive performance of our proposal. Our method inspires researchers to find a trade-off that simultaneously leverages the geometry of the Wasserstein distance and the favorable unbiased gradient estimate property of MMD while designing new distributional RL algorithms in the future.

## 2 PRELIMINARY KNOWLEDGE

### 2.1 DISTRIBUTIONAL REINFORCEMENT LEARNING

In classical RL, an agent interacts with an environment via a Markov decision process (MDP), a 5-tuple $(\mathcal{S}, \mathcal{A}, R, P, \gamma)$, where $\mathcal{S}$ and $\mathcal{A}$ are the state and action spaces, respectively. $P$ is the environment transition dynamics, $R$ is the reward function and $\gamma \in (0, 1)$ is the discount factor.

**From Value function to Value distribution.** Given a policy $\pi$, the discounted sum of future rewards is a random variable $Z^\pi(s, a) = \sum_{t=0}^{\infty} \gamma^t R(s_t, a_t)$, where $s_0 = s$, $a_0 = a$, $s_{t+1} \sim P(\cdot|s_t, a_t)$, and $a_t \sim \pi(\cdot|s_t)$. In the control setting, expectation-based RL is based on the action-value function $Q^\pi(s, a)$, which is the expectation of $Z^\pi(s, a)$, i.e., $Q^\pi(s, a) = \mathbb{E}[Z^\pi(s, a)]$. By contrast, distributional RL focuses on the action-value distribution, the full distribution of $Z^\pi(s, a)$. The incorporation of additional distributional knowledge intuitively interprets its empirical success.

**Distributional Bellman operators.** For the policy evaluation in expectation-based RL, the action-value function is updated via Bellman operator $\mathcal{T}^\pi Q(s, a) = \mathbb{E}[R(s, a)] + \gamma \mathbb{E}_{s' \sim p, \pi}[Q(s', a')]$. In distributional RL, the distribution of $Z^\pi(s, a)$ is updated via the distributional Bellman operator $\mathfrak{T}^\pi$

$$\mathfrak{T}^\pi Z(s, a) :\overset{D}{=} R(s, a) + \gamma Z(s', a'), \tag{1}$$

where $s' \sim P(\cdot|s, a)$ and $a' \sim \pi(\cdot|s')$. The equality implies random variables of both sides are equal in distribution. The distributional Bellman operator $\mathfrak{T}^\pi$ is contractive under certain distribution divergence metrics. We provide a detailed discussion about more related works in Appendix A.

### 2.2 DIVERGENCES BETWEEN MEASURES

**Optimal Transport (OT) and Wasserstein Distance.** The optimal transport (OT) metric between two probability measures $(\mu, \nu)$ is defined as the solution of the linear program:

$$\min_{\Pi \in \mathbf{\Pi}(\mu, \nu)} \int c(x, y) \mathrm{d}\Pi(x, y), \tag{2}$$

where $c$ is the cost function and $\Pi$ is the joint distribution with marginals $(\mu, \nu)$. Wasserstein distance (a.k.a. earth mover distance) is a special case of optimal transport with the Euclidean norm as the cost function. In particular, given two scalar random variables $X$ and $Y$, $p$-Wasserstein metric $W_p$ between the distributions of $X$ and $Y$ can be simplified as

$$W_p(X, Y) = \left( \int_0^1 \left| F_X^{-1}(\omega) - F_Y^{-1}(\omega) \right|^p d\omega \right)^{1/p}, \tag{3}$$

where $F^{-1}$ is the inverse cumulative distribution function of a random variable. The desirable geometric property of Wasserstein distance allows it to recover full support of measures, but it suffers from the curse of dimension (Genevay et al., 2019; Arjovsky et al., 2017).

**Maximum Mean Discrepancy.** The squared Maximum Mean Discrepancy (MMD) $\mathrm{MMD}_k^2$ with the kernel $k$ is formulated as

$$\mathrm{MMD}_k^2 = \mathbb{E}[k(X, X')] + \mathbb{E}[k(Y, Y')] - 2\mathbb{E}[k(X, Y)], \tag{4}$$

where $k(\cdot, \cdot)$ is a continuous kernel on $\mathcal{X}$. $X'$ (resp. $Y'$) is a random variable independent of $X$ (resp. $Y$). If $k$ is a trivial kernel, MMD degenerates to the energy distance. Mathematically, the "flat" geometry that MMD induces on the space of probability measures does not faithfully lift the ground distance (Feydy et al., 2019), but MMD is cheaper to compute than OT and has a smaller sample complexity, i.e., approximating the distance with samples of measures (Genevay et al., 2019). We provide the detailed introduction of more distribution divergences in Appendix B.

## 3 ROLES OF DISTRIBUTION DIVERGENCE AND REPRESENTATION

### 3.1 DISTRIBUTIONAL RL: FROM NEURAL FITTED Q TO Z ITERATION

**Neural Fitted Q-Iteration.** It is known that Deep Q Networks (Mnih et al., 2015) can be simplified into *Neural Fitted Q-Iteration* (Fan et al., 2020) under tricks of experience replay and the target network, where we update $Q_\theta(s, a)$ parameterized by $\theta$ in each iteration $k$:

$$Q_\theta^{k+1} = \underset{Q_\theta}{\operatorname{argmin}} \frac{1}{n} \sum_{i=1}^{n} \left[ y_i - Q_\theta^k(s_i, a_i) \right]^2,$$ (5)

where the target $y_i = r(s_i, a_i) + \gamma \max_{a \in \mathcal{A}} Q_{\theta^*}^k(s_i', a)$ is fixed within every $T_{\text{target}}$ steps to update target network $Q_{\theta^*}$ with parameters $\theta^*$ by letting $\theta^* = \theta$ and the experience buffer induces independent samples $\{(s_i, a_i, r_i, s_i')\}_{i \in [n]}$. In an ideal case that neglects the non-convexity and TD approximation errors, we have $Q_\theta^{k+1} = \mathcal{T} Q_\theta^k$, which is exactly equivalent to the updating rule under Bellman optimality operator.

**Neural Fitted Z-Iteration.** Analogous to Neural Fitted Q-iteration, we can also simplify value-based distributional RL methods based on a parameterized $Z_\theta$ into a *Neural Fitted Z-Iteration* as

$$Z_\theta^{k+1} = \underset{Z_\theta}{\operatorname{argmin}} \frac{1}{n} \sum_{i=1}^{n} d_p(Y_i, Z_\theta^k(s_i, a_i)),$$ (6)

where the target $Y_i = R(s_i, a_i) + \gamma Z_{\theta^*}^k(s_i', \pi_Z(s'))$ with $\pi_Z(s') = \operatorname{argmax}_{a'} \mathbb{E}\left[ Z_{\theta^*}^k(s', a') \right]$ is fixed within every $T_{\text{target}}$ steps to update target network $Z_{\theta^*}$, and $d_p$ is the distribution divergence.

### 3.2 KEY ROLES OF $d_p$ AND $Z_\theta$

Within the Neural Fitted Z-Iteration framework proposed in Eq. 6, we observe that the choice of representation manner on $Z_\theta$ and the metric $d_p$ are pivotal for the distributional RL algorithms. For instance, QR-DQN (Dabney et al., 2018b) approximates Wasserstein distance $W_p$, which leverages quantiles to represent the distribution of $Z_\theta$. C51 (Bellemare et al., 2017a) represents $Z_\theta$ via a categorical distribution under the convergence of Cramér distance (Bellemare et al., 2017b; Rowland et al., 2018), while MMD distributional RL (MMDDRL) (Nguyen et al., 2020) learns samples to represent the distribution of $Z_\theta$ based on MMD. We compare characteristics of these distribution divergence, including the convergence rate and sample complexity, in Table 1. Theoretical results regarding Sinkhorn divergence is based on (Genevay et al., 2019) and the detailed convergence proof of other distances is also provided in Appendix B. In summary, we argue that $d_p$ and $Z_\theta$ are two crucial factors in distributional RL design, based on which we introduce Sinkhorn distributional RL.

| Algorithm | $d_p$ **Distribution Divergence** | **Representation** $Z_\theta$ | **Convergence Rate of** $\mathfrak{T}^\pi$ | **Sample Complexity of** $d_p$ |
|---|---|---|---|---|
| C51 | Cramér distance | Histogram | $\sqrt{\gamma}$ | |
| QR-DQN | Wasserstein distance | Quantiles | $\gamma$ | $\mathcal{O}(n^{-\frac{1}{d}})$ |
| MMDDRL | MMD | Samples | $\gamma^{\alpha/2}$ with kernel $k_\alpha$ | $\mathcal{O}(1/n)$ |
| SinkhornDRL (ours) | Sinkhorn divergence | Samples | $\gamma$ ($\varepsilon \to 0$) $\gamma^{\alpha/2}$ ($\varepsilon \to \infty$) | $\mathcal{O}(n^{\frac{\kappa}{\varepsilon \lfloor d/2 \rfloor \sqrt{n}}})$ ($\varepsilon \to 0$) $\mathcal{O}(n^{-\frac{1}{2}})$ ($\varepsilon \to \infty$) |

Table 1: Comparison between typical distributional RL algorithms under different distribution divergences and represention of $Z_\theta$. $k_\alpha = -\|x - y\|^\alpha$ in MMDDRL, $d$ is the sample dimension and $\kappa = 2\beta d + \|c\|_\infty$, where the cost function $c$ is $\beta$-Lipschitz (Genevay et al., 2019). Sample complexity of MMD can be improved to $\mathcal{O}(1/n)$ using kernel herding technique (Chen et al., 2012).

# 4 SINKHORN DISTRIBUTIONAL RL (SINKHORNDRL)

In this section, we firstly introduce Sinkhorn divergence and apply it in distributional RL. Next, we conduct a theoretical analysis about the convergence and a new moment matching manner of our algorithm under the Sinkhorn divergence. Finally, a practical Sinkhorn iteration algorithm is introduced to evaluate the Sinkhorn divergence.

## 4.1 SINKHORN DIVERGENCE AND GENETIC ALGORITHM

We design Sinkhorn distributional RL algorithm via Sinkhorn divergence. Sinkhorn divergence (Sinkhorn, 1967) is a tractable loss to approximate the optimal transport problem by leveraging an entropic regularization to turn the original Wasserstein distance into a differentiable and more robust quantity. The resulting loss can be computed using Sinkhorn fixed point iterations, which is naturally suitable for modern deep learning frameworks. In particular, the entropic smoothing generates a family of losses interpolating between MMD. As such, it allows us to find a sweet trade-off that simultaneously leverages the geometry of Wasserstein distance on the one hand, and the favorable high-dimensional sample complexity and unbiased gradient estimates of MMD. We introduce the entropic regularized Wassertein distance $\mathcal{W}_{c,\varepsilon}(\mu, \nu)$ as

$$\min_{\Pi \in \mathbf{\Pi}(\mu, \nu)} \int c(x, y) \mathrm{d}\Pi(x, y) + \varepsilon \mathrm{KL}(\Pi | \mu \otimes \nu), \tag{7}$$

where $\mathrm{KL}(\Pi | \mu \otimes \nu) = \int \log \left( \frac{\Pi(x,y)}{\mathrm{d}\mu(x)\mathrm{d}\nu(y)} \right) \mathrm{d}\Pi(x, y)$ is a strongly convex regularization. The impact of this entropy regularization is similar to $\ell_2$ ridge regularization in linear regression. Next, the sinkhorn loss (Feydy et al., 2019; Genevay et al., 2018) between two measures $\mu$ and $\nu$ is defined as

$$\overline{\mathcal{W}}_{c,\varepsilon}(\mu, \nu) = 2\mathcal{W}_{c,\varepsilon}(\mu, \nu) - \mathcal{W}_{c,\varepsilon}(\mu, \mu) - \mathcal{W}_{c,\varepsilon}(\nu, \nu). \tag{8}$$

As demonstrated by (Feydy et al., 2019), the Sinkhorn divergence $\overline{\mathcal{W}}_{c,\varepsilon}(\mu, \nu)$ is convex, smooth and positive definite that metrizes the convergence in law. In statistical physics, $\mathcal{W}_{c,\varepsilon}(\mu, \nu)$ can be re-factored as a projection problem:

$$\mathcal{W}_{c,\varepsilon}(\mu, \nu) := \min_{\Pi \in \mathbf{\Pi}(\mu, \nu)} \mathrm{KL}\left(\Pi | \mathcal{K}\right), \tag{9}$$

where $\mathcal{K}$ is the Gibbs distribution and its density function satisfies $d\mathcal{K}(x, y) = e^{-\frac{c(x,y)}{\varepsilon}} d\mu(x) d\nu(y)$. This problem is often referred to as the "static Schrödinger problem" (Léonard, 2013; Rüschendorf & Thomsen, 1998) as it was initially considered in statistical physics.

**Distributional RL with Sinkhorn Divergence and Particle Representation.** The key of applying Sinkhorn divergence in distributional RL is to simply leverage the Sinkhorn loss $\overline{\mathcal{W}}_{c,\varepsilon}$ to measure the distance between the current action-value distribution $Z_\theta(s, a)$ and the target distribution $\mathfrak{T}^\pi Z_\theta(s, a)$, yielding $\overline{\mathcal{W}}_{c,\varepsilon}(Z_\theta(s, a), \mathfrak{T}^\pi Z_\theta(s, a))$ for each $s, a$ pairs. In terms of the representation for $Z_\theta(s, a)$, we employ the unrestricted statistics, i.e., deterministic samples, due to its superiority in MMDDRL (Nguyen et al., 2020), instead of using predefined statistic functionals, e.g., quantiles in QR-DQN (Dabney et al., 2018b) or categorical distribution in C51 (Bellemare et al., 2017a).

---

**Algorithm 1** Generic Sinkhorn distributional RL Update

---

**Require**: Number of generated samples $N$, the cost function $c$ and hyperparameter $\varepsilon$.
**Input**: Sample transition $(s, a, r', s')$
 1: **if** Policy evaluation **then**
 2:    $a^* \sim \pi(\cdot | s')$.
 3: **else**
 4:    $a^* \leftarrow \arg\max_{a' \in \mathcal{A}} \frac{1}{N} \sum_{i=1}^{N} Z_\theta\left(s', a'\right)_i$
 5: **end if**
 6:  $\mathfrak{T}Z_i \leftarrow r + \gamma Z_{\theta^*}\left(s', a^*\right)_i, \forall 1 \le i \le N$
**Output**: $\overline{\mathcal{W}}_{c,\varepsilon}\left(\{Z_\theta(s, a)_i\}_{i=1}^{N}, \{\mathfrak{T}Z_\theta(s, a)_j\}_{j=1}^{N}\right)$

---

More concretely, we use neural networks to generate samples that approximate the value distribution. This can be expressed as $Z_\theta(s,a) := \{Z_\theta(s,a)_i\}_{i=1}^N$, where $N$ is the number of generated samples. We refer to the samples $\{Z_\theta(s,a)_i\}_{i=1}^N$ as *particles*. Then we leverage the Dirac mixture $\frac{1}{N}\sum_{i=1}^N \delta_{Z_\theta(s,a)_i}$ to approximate the true density function of $Z^\pi(s,a)$, thus minimizing the Sinkhorn divergence between the approximate distribution and its distributional Bellman target. A detailed and generic distributional RL algorithm with Sinkhorn divergence and particle representation is provided in Algorithm 1.

**Remark.** From the general algorithm framework in Algorithm 1, our SinkhornDRL generally modifies the distribution divergence comparing the state-of-the-art MMDDRL (Nguyen et al., 2020), but SinkhornDRL fundamentally falls into Wasserstein distance-based distributional RL family as discussed in Appendix A. As such, QR-DQN and MMDDRL are direct counterparts for Sinkhorn-DRL, and the follow-up works IQN (Dabney et al., 2018a) and FQF (Yang et al., 2019) can naturally extend both MMDDRL and SinkhornDRL as discussed in (Nguyen et al., 2020).

## 4.2 THEORETICAL ANALYSIS UNDER SINKHORN DIVERGENCE

**Convergence.** Firstly, we denote the supreme form of Sinkhorn divergence as $\overline{\mathcal{W}}_{c,\varepsilon}^\infty(\mu,\nu)$:

$$\overline{\mathcal{W}}_{c,\varepsilon}^\infty(\mu,\nu) = \sup_{(x,a)\in\mathcal{S}\times\mathcal{A}} \overline{\mathcal{W}}_{c,\varepsilon}(\mu(x,a),\nu(x,a)). \tag{10}$$

We will use $\overline{\mathcal{W}}_{c,\varepsilon}^\infty(\mu,\nu)$ to establish the convergence of $\mathfrak{T}^\pi$ in Theorem 1.

**Theorem 1.** *If we leverage Sinkhorn loss $\overline{\mathcal{W}}_{c,\varepsilon}(\mu,\nu)$ in Eq. 8 as the distribution divergence in distributional RL, and **choose the unrectified kernel** $k_\alpha := -\|x-y\|^\alpha$ **as** $-c$ ($\alpha > 0$), it holds that*

*(1) ($\varepsilon \to 0$) $\overline{\mathcal{W}}_{c,\varepsilon}(\mu,\nu) \to 2W_\alpha(\mu,\nu)$. When $\varepsilon = 0$, $\mathfrak{T}^\pi$ is a $\gamma$-contraction under $\overline{\mathcal{W}}_{c,\varepsilon}^\infty$.*

*(2) ($\varepsilon \to +\infty$) $\overline{\mathcal{W}}_{c,\varepsilon}(\mu,\nu) \to MMD_{k_\alpha}^2(\mu,\nu)$. When $\varepsilon = +\infty$, $\mathfrak{T}^\pi$ is $\gamma^{\alpha/2}$-contractive under $\overline{\mathcal{W}}_{c,\varepsilon}^\infty$.*

*(3) ($\varepsilon \in (0,+\infty)$), $\mathfrak{T}^\pi$ is a **contractive** operator under $\overline{\mathcal{W}}_{c,\varepsilon}^\infty$. The related non-constant contraction factor $\Delta(\gamma,\alpha) < 1$ also depends on the distribution sequence in distributional Bellman iterations.*

We provide the long yet rigorous proof of Theorem 1 in Appendix C. Theorem 1 (1) and (2) are follow-up conclusions in terms of the convergence behavior of $\mathfrak{T}^\pi$ based on the interpolation relationship between Sinkhorn divergence with Wasserstein distance and MMD (Genevay et al., 2018). Our key theoretical contribution is for the general $\varepsilon \in (0,\infty)$, in which we conclude that $\mathfrak{T}^\pi$ is a contractive operator. The crux of the proof is two-fold. Firstly, we show the a variant of scale sensitive property of Sinkhorn divergence when $c = -\kappa_\alpha$, where the resulting non-constant scaling factor is also determined by the specified two probability measures. Next, we propose a new distribution Contraction mapping theorem in Theorem 2 of Appendix C, based on which we eventually arrive at the convergence of distributional Bellman operator under $\overline{\mathcal{W}}_{c,\varepsilon}^\infty$. Intriguingly yet reasonably, the contraction factor $\Delta(\gamma,\alpha)$ is non-constant but a function less than 1 that also depends on the distribution sequence while iteratively applying distribution Bellman updates. Our non-trivial proof about Sinkhorn divergence can even contribute to the optimal transport literature.

**Consistency with Related Conclusions.** As Sinkhorn divergence interpolates between Wasserstein distance and MMD, its contraction property when the cost function holds $c = -k_\alpha$ for the general $\varepsilon \in [0,\infty]$ is intuitive. Note that if we choose Gaussian kernels as the cost function, there will be no concise and consistent contraction results as Theorem 1 (3). This conclusion is also consistent with MMDDRL (Nguyen et al., 2020), where $\mathfrak{T}^\pi$ is generally not a contraction operator under MMD equipped with Gaussian kernels as a counterexample has been pointed out in MMDDRL (when $\varepsilon \to +\infty$). To be consistent with the contraction property analyzed in our theory (Theorem 1 (3)), we employ the rectified kernel $k_\alpha$ as the cost function in our experiments and set $\alpha = 2$, under which SinkhornDRL suggests a favorable performance in Section 5.

**Regularized Moment Matching under Sinkhorn Divergence Associated with Gaussian Kernels.** We further examine the potential connection between SinkhornDRL with existing distributional RL families. Inspired by the similar manner in MMDDRL (Nguyen et al., 2020), we find that the Sinkhorn divergence with the Gaussian kernel can also promote to match all moments be-

tween two distributions. More specifically, the Sinkhorn divergence can be rewritten as a regularized moment matching form in Proposition 1.

**Proposition 1.** *For* $\varepsilon \in (0, +\infty)$*, Sinkhorn divergence* $\overline{\mathcal{W}}_{c,\varepsilon}(\mu, \nu)$ *associated with Gaussian kernels* $k(x, y) = \exp(-(x - y)^2/(2\sigma^2))$ *as* $-c$*, is equivalent to*

$$\overline{\mathcal{W}}_{c,\varepsilon}(\mu, \nu) := \sum_{n=0}^{\infty} \frac{1}{\sigma^{2n} n!} \left( \tilde{M}_n(\mu) - \tilde{M}_n(\nu) \right)^2 + \varepsilon \mathbb{E} \left[ \log \frac{(\Pi_{\varepsilon}^*(X, Y))^2}{\Pi_{\varepsilon}^*(X, X')\Pi_{\varepsilon}^*(Y, Y')} \right], \qquad (11)$$

*where* $\Pi_{\varepsilon}^*$ *denotes the optimal* $\Pi$ *determined by* $\varepsilon$ *by evaluating the Sinkhorn divergence via* $\min_{\Pi \in \Pi(\mu, \nu)} \overline{\mathcal{W}}_{c,\varepsilon}(\mu, \nu)$. $\tilde{M}_n(\mu) = \mathbb{E}_{x \sim \mu} \left[ e^{-x^2/(2\sigma^2)} x^n \right]$*, and similarly for* $\tilde{M}_n(\nu)$*.*

We provide the proof of Proposition 1 in Appendix D. Similar to MMDDRL associated with a Gaussian kernel (Nguyen et al., 2020), Sinkhorn divergence approximately performs a regularized moment matching scaled by $e^{-x^2/(2\sigma^2)}$.

**Equivalence to Regularized MMD Distributional RL.** Based on Proposition 1, we can immediately establish the connection between Sinkhorn divergence and MMD in Corollary 1, indicating that minimizing Sinkhorn divergence between two distributions is equivalent to minimizing a regularized squared MMD.

**Corollary 1.** *For* $\varepsilon \in (0, +\infty)$ *and denote* $\Pi_{\varepsilon}^*$ *as the optimal* $\Pi$ *by evaluating the Sinkhorn divergence, it holds that*

$$\overline{\mathcal{W}}_{c,\varepsilon} := MMD^2_{-c}(\mu, \nu) + \varepsilon \mathbb{E} \left[ \log \frac{(\Pi_{\varepsilon}^*(X, Y))^2}{\Pi_{\varepsilon}^*(X, X')\Pi_{\varepsilon}^*(Y, Y')} \right], \qquad (12)$$

*where we use* $\overline{\mathcal{W}}_{c,\varepsilon}$ *to replace* $\overline{\mathcal{W}}_{c,\varepsilon}(\mu, \nu)$ *for short.*

Proof of Corollary 1 is provided in Appendix D. It is worthy of noting that this equivalence is established for the general case when $\varepsilon \in (0, +\infty)$, and it does not hold in the limit cases when $\varepsilon \to 0$ or $+\infty$. For example, when $\varepsilon \to +\infty$, the second part including $\varepsilon$ in Eq. 12 is not expected to dominate. This is owing to the fact that the regularization term would be 0 as $\Pi_{\varepsilon}^* \to \mu \otimes \nu$ when $\varepsilon \to +\infty$. In summary, even though the Sinkhorn divergence was initially proposed to serve as an entropy regularized Wasserterin distance when the cost function $c = \kappa_{\alpha}$, it turns out that it is equivalent to a regularized MMD if associated with Gaussian kernels, as revealed in Corollary 1.

### 4.3 DISTRIBUTIONAL RL VIA SINKHORN ITERATIONS

The theoretical analysis in Section 4.2 sheds light on the behavior of distributional RL with Sinkhorn divergence, but another crucial issue we need to address is how to evaluate the Sinkhorn loss effectively. Due to the advantages of Sinkhorn divergence that both enjoys geometry property of optimal transport and the computational effectiveness of MMD, we can utilize Sinkhorn's algorithm, i.e., Sinkhorn Iterations (Sinkhorn, 1967; Genevay et al., 2018), to evaluate the Sinkhorn loss. Notably, Sinkhorn iteration with $L$ steps yields a differentiable and solvable efficiently loss function as the main burden involved in it is the matrix-vector multiplication, which streams well on the GPU with simply adding extra differentiable layers on the typical deep neural network, such as a DQN architecture.

Specifically, given two sample sequences $\{Z_i\}_{i=1}^N, \{\mathfrak{T}Z_j\}_{j=1}^N$ in the distributional RL algorithm, the optimal transport distance is equivalent to the form:

$$\min_{P \in \mathbb{R}_+^{N \times N}} \left\{ \langle P, \hat{c} \rangle; P\mathbf{1}_N = \mathbf{1}_N, P^{\top}\mathbf{1}_N = \mathbf{1}_N \right\}, \qquad (13)$$

where the empirical cost function $\hat{c}_{i,j} = c(Z_i, \mathfrak{T}Z_j)$. By adding entropic regularization on optimal transport distance, Sinkhorn divergence can be viewed to restrict the search space of $P$ in the following scaling form:

$$P_{i,j} = a_i \mathcal{K}_{i,j} b_j, \qquad (14)$$

where $\mathcal{K}_{i,j} = e^{-\hat{c}_{i,j}/\varepsilon}$ is the Gibbs kernel defined in Eq. 9. This allows us to leverage iterations regarding the vectors $a$ and $b$. More specifically, we initialize $b_0 = \mathbf{1}_N$, and then the Sinkhorn

---

**Algorithm 2** Sinkhorn Iterations to Approximate $\overline{\mathcal{W}}_{c,\varepsilon}\left(\{Z_i\}_{i=1}^N, \{\mathfrak{T}Z_j\}_{j=1}^N\right)$

---

**Input**: Two samples sequences $\{Z_i\}_{i=1}^N, \{\mathfrak{T}Z_j\}_{j=1}^N$, number of Sinkhorn iterations $L$ and hyperparameter $\varepsilon$.

1:  $\hat{c}_{i,j} = c(Z_i, \mathfrak{T}Z_j)$ for $\forall i = 1, ..., N, j = 1, ..., N$
2:  $\mathcal{K}_{i,j} = \exp(-\hat{c}_{i,j}/\varepsilon)$
3:  $b_0 \leftarrow \mathbf{1}_N$
4:  **for** $l = 1, 2, ..., L$ **do**
5:      $a_l \leftarrow \frac{\mathbf{1}_N}{\mathcal{K}b_{l-1}}, b_l \leftarrow \frac{\mathbf{1}_N}{\mathcal{K}a_l}$
6:  **end for**
7:  $\widehat{\overline{\mathcal{W}}}_{c,\varepsilon}\left(\{Z_i\}_{i=1}^N, \{\mathfrak{T}Z_j\}_{j=1}^N\right) = \langle (K \odot \hat{c})b, a \rangle$

**Return**: $\widehat{\overline{\mathcal{W}}}_{c,\varepsilon}\left(\{Z_i\}_{i=1}^N, \{\mathfrak{T}Z_j\}_{j=1}^N\right)$

---

iterations are expressed as

$$a_{l+1} \leftarrow \frac{\mathbf{1}_N}{\mathcal{K}b_l} \quad \text{and} \quad b_{l+1} \leftarrow \frac{\mathbf{1}_N}{\mathcal{K}^\top a_{l+1}}, \tag{15}$$

where $\dot{\div}$ indicates an entry-wise division. It has been proven that Sinkhorn iteration asymptotically converges to the true loss in a linear rate (Genevay et al., 2018; Franklin & Lorenz, 1989; Cuturi, 2013; Jason Altschuler, 2017). We provide a detailed algorithm description of Sinkhorn iterations in Algorithm 2. With the efficient and differential Sinkhorn iterations, we can easily evaluate the Sinkhorn divergence and thus let our algorithm enjoy its theoretical advantages. In practice, we need to choose $L$ and $\varepsilon$, and we conduct a rigorous sensitivity analysis in Section 5.

## 5 EXPERIMENTS

We demonstrate the effectiveness of SinkhornDRL as described in Algorithm 1 on the full 55 Atari 2600 games. Specifically, we leverage the same architecture as QR-DQN (Dabney et al., 2018b), and replace the quantiles output with $N$ particles, i.e., samples. In contrast to MMDDRL, SinkhornDRL only changes the distribution divergence from MMD to Sinkhorn divergence, and therefore the potential superiority in the performance can be attributed to the advantages of Sinkhorn divergence.

**Baselines.** Due to the interpolation feature of Sinkhorn divergence between Wassertein distance and MMDDRL, we choose three typical distributional RL algorithms as classic baselines, including QR-DQN (Dabney et al., 2018b) that approximates the Wasserstein distance, C51 (Bellemare et al., 2017a) and MMDDRL (Nguyen et al., 2020), as well as DQN (Mnih et al., 2015). MMDDRL algorithm is implemented with the same architecture as QRDQN, and leverages Gaussian kernels $k_h(x, y) = \exp(-(x-y)^2/h)$ with the kernel mixture trick covering a range of bandwidths $h$, which is same as the basic setting in the original MMDDQN paper (Nguyen et al., 2020). We deploy all algorithms on 55 Atari 2600 games, and reported results are averaged over 3 seeds with the shade indicating the standard deviation. We runs 10M time steps (40M frames) for the computation cost reason, but we report learning curves across all games to make results convincing enough.

**Hyperparameter settings.** For a fair comparison with QR-DQN, C51 and MMDDRL, we used the same hyperparamters: the number of generated samples $N = 200$, Adam optimizer with lr $= 0.00005, \epsilon_{\text{Adam}} = 0.01/32$. We used a target network to compute the distributional Bellman target, which fits well in the Neural Fitted Z-Iteration framework. In addition, we choose number of Sinkhorn iterations $L = 10$ and smoothing hyperparameter $\varepsilon = 10.0$ in Section 5.1 as they are not sensitive within a proper interval as demonstrated in Section 5.2. We choose the unrectified kernel as the cost function, i.e.,$-c = k_\alpha$, and select $\alpha = 2$ in $k_\alpha$ in our SinkhornDRL algorithm.

### 5.1 PERFORMANCE OF SINKHORNDRL

Figure 1 illustrates that SinkhornDRL can achieve the competitive performance across 55 Atari games compared with various baseline algorithms with different metrics $d_p$ and representation manners on $Z_\theta$. On a large number of games, e.g., Tennis, Seaquest and Atlantis, SinkhornDRL can

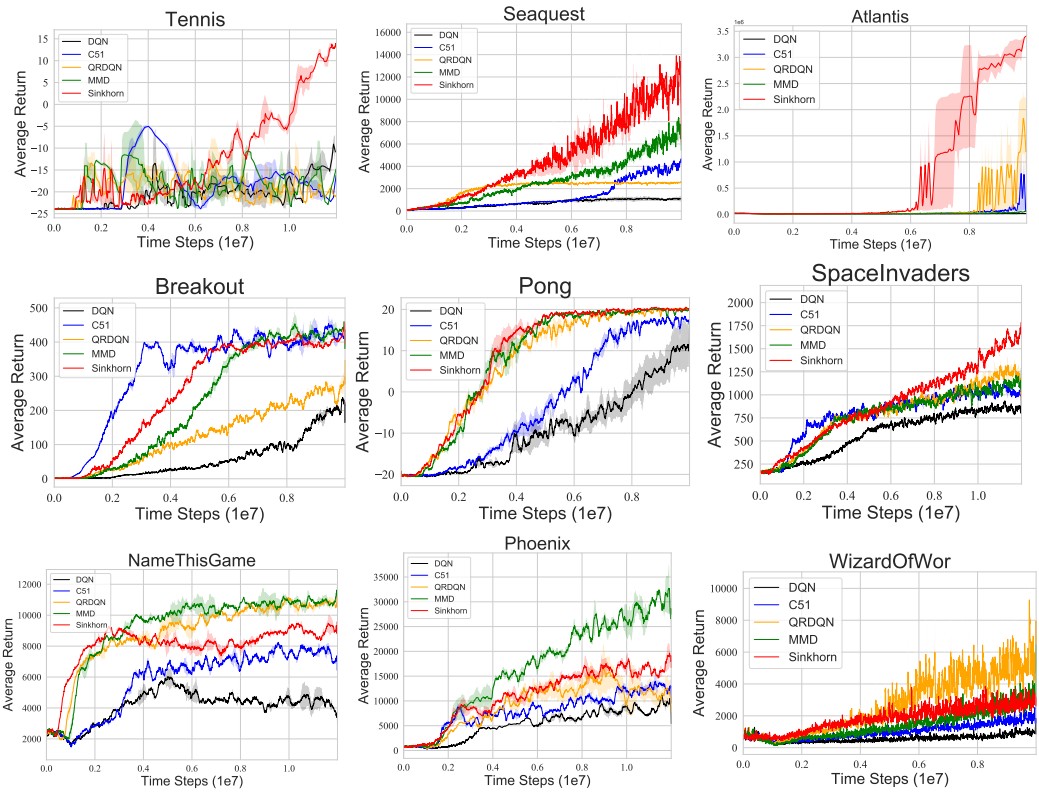

Figure 1: Learning curves of SinkhornDRL algorithm compared with DQN, C51, QR-DQN and MMD, on nine typical Atari games over 3 seeds.

significantly outperform other baselines, especially on Tennis where other algorithms even fail to converge. The improvement of SinkhornDRL over MMDDRL empirically verifies the regularization advantage of the Sinkhorn as analyzed in Corollary 1. On some games, e.g., Breakout, Pong and SpaceInvaders, SinkhornDRL is on par with MMDDRL and other baselines, while on the last row in Figure 1, SinkhornDRL is slightly inferior to the state-of-the-art algorithm. We provide learning curves of all typical distributional RL algorithms on all 55 Atari games in Appendix F, where SinkhornDRL still achieves the competitive performance in general.

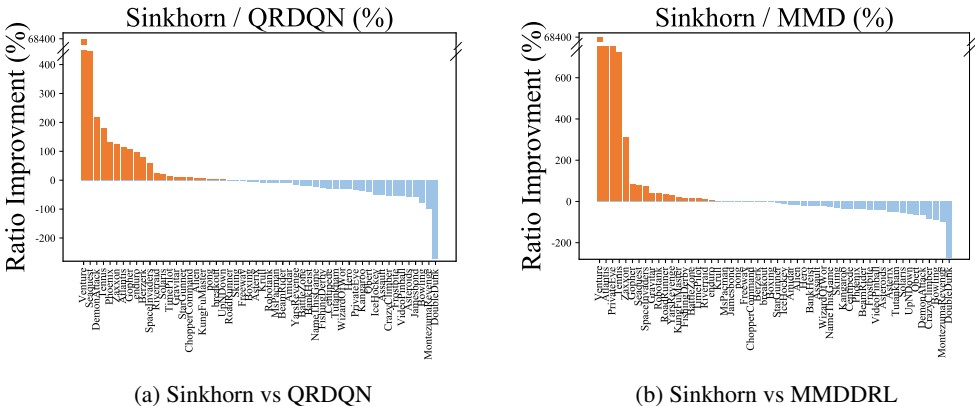

(a) Sinkhorn vs QRDQN

(b) Sinkhorn vs MMDDRL

Figure 2: Ratio improvement of return for Sinkhorn distributional RL algorithm over QRDQN (left) and MMDDRL (right) over 3 seeds. For example, the ratio improvement is calculated by (Sinkhorn - QRDQN) / QRDQN in the left.

We conduct a ratio improvement comparison across 55 Atari games between SinkhornDRL with QRDQN and MMDDRL, respectively. Figure 2 showcases that by comparing with QRDQN (left), SinkhornDRL achieves better performance across almost half of considered games and the superiority of SinkhornDRL is significant across a large amount of games, including Venture, Seaquest, Tennis and Phoenix. This empirical outperformance verifies the effectiveness of smoothing Wasserstein distance in distributional RL. In contrast with MMDDRL, the advantage of SinkhornDRL is reduced with the performance improvement on a smaller proportion of games, but a remarkable performance improvement for SinkhornDRL on a large amount of games can be easily observed. We also report mean and median of best human-normalized scores in Table 2 of Appendix E, where SinkhornDRL achieves almost state-of-the-art performance as MMDDRL on average.

Therefore, we conclude that SinkhornDRL is competitive with the state-of-the-art distributional RL algorithms, e.g., MMDDRL, and can be extremely superior over existing algorithms on a large proportion of games. This empirical success can be owing to theoretical advantage of Sinkhorn divergence that simultaneously makes full use of the data geometry from Wasserstein distance and the unbiased gradient estimate property from MMD, which coincides with results in Theorem 1.

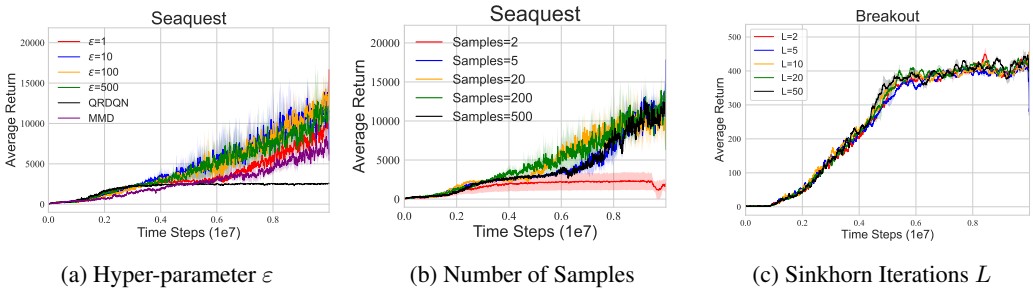

(a) Hyper-parameter $\varepsilon$       (b) Number of Samples       (c) Sinkhorn Iterations $L$

Figure 3: Sensitivity analysis of SinkhornDRL on Breakout regarding $\varepsilon$, number of samples, and number of iteration $L$. Learning curves are reported over 3 seeds.

## 5.2 SENSITIVITY ANALYSIS AND COMPUTATIONAL COST

The limit behavior connection in Theorem1 (1) and (2) between SinkhornDRL with QR-DQN and MMDDRL may not be rigorously verified in numerical experiments as an overly large or small $\varepsilon$ will lead to numerical instability of Sinkhorn iterations in Algorithm 2, worsening its performance, as shown in Figure 3 (a). In practice, we choose a proper $\varepsilon = 10$ across all games. SinkhornDRL also requires a proper number of iterations $L$ and samples $N$. For example, a small $N$, e.g., $N = 2$ in Seaquest in Figure 3 (b) leads to the divergence of algorithms, while an overly large $N$ can degrade the performance and meanwhile increases the computational burden (Appendix G). We conjecture that using larger networks to represent more samples is more likely to suffer from the overfitting issue, yielding the instability in the RL training (Bjorck et al., 2021). Therefore, we choose $N = 200$ to attain an appealing performance with the computational effectiveness. For the computation cost (Appendix G), SinkhornDRL increases around $50\%$ computation cost compared with QR-DQN and C51, but only slightly increases the overhead (by around $20\%$) in contrast to MMDDRL. Please refer to Appendix G for more detailed results and discussion.

## 6 DISCUSSIONS AND CONCLUSION

To extend our algorithm for better performance, implicit generative models, including parameterizing the cost function in Sinkhorn loss, can be further incorporated. We leave it as the future work. Moreover, other divergences, e.g., those that can also smooth Wasserstein distance, can also be applied into the design of distributional RL algorithms in the future.

In this paper, a novel family of distributional RL algorithms based on Sinkhorn divergence is proposed that accomplishes a competitive performance compared with the-state-of-the-art distributional RL algorithms on 55 Atari games. Theoretical analysis about the convergence and moment matching behavior is provided along with a rigorous empirical verification. Sinkhorn distributional RL will lead to an important contribution among the research community.

**Ethics Statement.** Our study is about the design of distributional RL algorithms, which is not involved with any ethics issue.

**Reproducibility Statement.** Our results is based on the public implementation released in (Zhang, 2018) with necessary implementation details given in Appendix F. We also provide the detailed proof from Appendix C to Appendix D.

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

## A  RELATED WORK

Based on the choice of distribution divergence and the distribution representation manner of $Z_\theta$, distributional RL algorithms can be mainly categorized into three classes, including **categorical, Wasserstein Distance and MMD distributional RL**. Finally, we discuss their relationships with our proposed SinkhornDRL.

**Categorical Distributional RL.**   As the first successful distributional RL family, categorical distributional RL (Bellemare et al., 2017a) represents the value distribution $\eta$ by the categorical distribution $\hat{\eta} = \sum_{i=1}^{N} p_i \delta_{z_i}$, where $\{z_i\}_{i=1}^{N}$ $(z_1 < ... < z_N)$ is the fixed supports within the pre-specified interval $[l, u]$ and $p_i$ is the approximated categorical probability in each bin, respectively. Within this algorithm family, C51 (Bellemare et al., 2017a) leverages a neural network to approximate the categorical probabilities $p_i$ and apply a projected KL divergence between the target and current categorical value distributions. C51 has also been shown the theoretical contraction under the Cramér distance (Bellemare et al., 2017b; Rowland et al., 2018), and empirically performs favorably in the suite of Atari games.

**Wasserstein Distance Distributional RL.**   As directly solving wasserstein distance in Eq. 16 is tricky, QR-DQN (Dabney et al., 2018b) firstly proposed to use quantile regression to approximate Wasserstein distance $W_p$. QR-DQN leverages quantiles to represent the distribution $\eta$ of $Z_\theta$, i.e., $\hat{\eta} = \sum_{i=1}^{N} \delta_{z_i}$, where $\{z_i\}_{i=1}^{N}$ is the learnable support atoms as the quantile values of a fixed quantile $\{\frac{2i-1}{2N}\}_{i=1}^{N}$. Implicit Quantile Networks (IQN) utilizes an implicit model to output quantile values $\{z_i\}_{i=1}^{N}$ more expressively, instead of the given ones in QRDQN. IQN also incorporates the risk measure in the framework of distributional RL. A follow-up work Fully parameterized Quantile Function (FQF) (Yang et al., 2019) improves IQN by proposing a more expressive quantile network, achieving better performance on Atari games. Non-crossing issue in quantile-regression has been raised and addressed properly in (Zhou et al., 2020) that further improves QR-DQN. The monotonic rational-quadratic splines are also used to learn smooth continuous quantile functions (Luo et al., 2021).

**MMD Distributional RL.**   MMD distributional RL (MMDDRL) (Nguyen et al., 2020) learns samples to represent the value distribution of $Z_\theta$ based on maximum mean discrepancy (MMD) in Eq. 20, achieving the state-of-the-art performance on Atari games.

**Discussion about SinkhornDRL.**   As a complementary Wasserstein distance-based distributional RL, our SinkhornDRL allows to solve Wasserstein Distance by incorporating the entropic regularization and circumstance the non-crossing issue in quantile regression intrinsically. Moreover, the cost function in SinkhornDRL can be further parameterized similar to IQN and FQF, which can intuitively achieve better performance. We leave the investigation towards this direction as future works. Meanwhile, SinkhornDRL is also closely linked with MMDDRL as Sinkhorn divergence interpolates between Wasserstein distance and MMD, and also learns the unrestricted statistics, i.e., samples, akin to MMDDRL.

## B  DEFINITION OF DISTANCES AND CONTRACTION

**Definition of distances.** Given two random variables $X$ and $Y$, $p$-Wasserstein metric $W_p$ between the distributions of $X$ and $Y$ is defined as

$$W_p(X, Y) = \left( \int_0^1 \left| F_X^{-1}(\omega) - F_Y^{-1}(\omega) \right|^p d\omega \right)^{1/p} = \| F_X^{-1} - F_Y^{-1} \|_p, \qquad (16)$$

which $F^{-1}$ is the inverse cumulative distribution function of a random variable with the cumulative distribution function as $F$. Further, $\ell_p$ distance (Elie & Arthur, 2020) is defined as

$$\ell_p(X, Y) := \left( \int_{-\infty}^{\infty} |F_X(\omega) - F_Y(\omega)|^p \, d\omega \right)^{1/p} = \| F_X - F_Y \|_p \qquad (17)$$

The $\ell_p$ distance and Wassertein metric are identical at $p = 1$, but are otherwise distinct. Note that when $p = 2$, $\ell_p$ distance is also called Cramér distance (Bellemare et al., 2017b) $d_C(X, Y)$. Also, the Cramér distance has a different representation given by

$$d_C(X, Y) = \mathbb{E}|X - Y| - \frac{1}{2}\mathbb{E}|X - X'| - \frac{1}{2}\mathbb{E}|Y - Y'|, \tag{18}$$

where $X'$ and $Y'$ are the i.i.d. copies of $X$ and $Y$. Energy distance (Székely, 2003; Ziel, 2020) is a natural extension of Cramér distance to the multivariate case, which is defined as

$$d_E(\mathbf{X}, \mathbf{Y}) = \mathbb{E}\|\mathbf{X} - \mathbf{Y}\| - \frac{1}{2}\mathbb{E}\|\mathbf{X} - \mathbf{X}'\| - \frac{1}{2}\mathbb{E}\|\mathbf{Y} - \mathbf{Y}'\|, \tag{19}$$

where $\mathbf{X}$ and $\mathbf{Y}$ are multivariate. Moreover, the energy distance is a special case of the maximum mean discrepancy (MMD), which is formulated as

$$\text{MMD}(\mathbf{X}, \mathbf{Y}; k) = \left(\mathbb{E}\left[k\left(\mathbf{X}, \mathbf{X}'\right)\right] + \mathbb{E}\left[k\left(\mathbf{Y}, \mathbf{Y}'\right)\right] - 2\mathbb{E}[k(\mathbf{X}, \mathbf{Y})]\right)^{1/2} \tag{20}$$

where $k(\cdot, \cdot)$ is a continuous kernel on $\mathcal{X}$. In particular, if $k$ is a trivial kernel, MMD degenerates to energy distance. Additionally, we further define the supreme MMD, which is a functional $\mathcal{P}(\mathcal{X})^{\mathcal{S} \times \mathcal{A}} \times \mathcal{P}(\mathcal{X})^{\mathcal{S} \times \mathcal{A}} \to \mathbb{R}$ defined as

$$\text{MMD}_\infty(\mu, \nu) = \sup_{(x,a) \in \mathcal{S} \times \mathcal{A}} \text{MMD}_\infty(\mu(x, a), \nu(x, a)) \tag{21}$$

We further present the convergence rate under different distribution divergences.

- $\mathcal{T}^\pi$ is $\gamma$-contractive under the supreme form of Wassertein distance $W_p$.
- $\mathcal{T}^\pi$ is $\gamma^{1/p}$-contractive under the supreme form of $\ell_p$ distance.
- $\mathcal{T}^\pi$ is $\gamma^{\alpha/2}$-contractive under $\text{MMD}_\infty$ with the kernel $k_\alpha(x, y) = -\|x - y\|^\alpha, \forall \alpha > 0$.

**Proof of Contraction.**

- Contraction under supreme form of Wasserstein diatance is provided in Lemma 3 (Bellemare et al., 2017a).
- Contraction under supreme form of $\ell_p$ distance can refer to Theorem 3.4 (Elie & Arthur, 2020).
- Contraction under $\text{MMD}_\infty$ is provided in Lemma 6 (Nguyen et al., 2020).

## C  PROOF OF THEOREM 1

*Proof.* **1.** $\varepsilon \to 0$ **and** $c = -k_\alpha$ It is obvious to observe that Sinkhorn loss degenerates to the wasserstein distance. We also have the conclusion that the distributional Bellman operator $\mathfrak{T}^\pi$ is $\gamma$-contractive under the supreme form of Wasserstein diatance, the proof of which is provided in Lemma 3 (Bellemare et al., 2017a). Since the above conclusion is made directly based on the limiting case when $\varepsilon = 0$, for an unspecified $\varepsilon$, we need a more rigorous proof. We show that their distance difference is **at most an infinitesimal** $\delta$.

Firstly, as $\mathcal{W}_{c,\varepsilon} \to W_\alpha$ and the regularization term is non-negative, using the language of $(\varepsilon, \delta)$ definition, we have: for $\forall \delta$, there exists a small positive constant $a$, such that $\mathcal{W}_{c,\varepsilon} - W_\alpha < \delta$ when $\epsilon \leq a$. Based on that, we have the contraction conclusion:

$$\overline{\mathcal{W}}_{-\kappa_\alpha, \varepsilon}^\infty(\mathfrak{T}^\pi Z_1, \mathfrak{T}^\pi Z_2) = \overline{\mathcal{W}}_{-\kappa_\alpha, \varepsilon}^\infty(\mathfrak{T}^\pi Z_1, \mathfrak{T}^\pi Z_2) - W_\alpha^\infty(\mathfrak{T}^\pi Z_1, \mathfrak{T}^\pi Z_2) + W_\alpha^\infty(\mathfrak{T}^\pi Z_1, \mathfrak{T}^\pi Z_2)$$
$$\leq \delta + W_\alpha^\infty(\mathfrak{T}^\pi Z_1, \mathfrak{T}^\pi Z_2), \tag{22}$$

where the second term $W_\alpha^\infty(\mathfrak{T}^\pi Z_1, \mathfrak{T}^\pi Z_2)$ is contractive, and thus for the unspecified $\varepsilon$, the only difference from the limting $\varepsilon = 0$ is an infinitesimal $\delta$, which will vanish as $\varepsilon \to 0$ or $a \to 0$.

**2.** $\varepsilon \to \infty$**.** Our complete proof is inspired by (Ramdas et al., 2017; Genevay et al., 2018). Recap the definition of squared MMD is

$$\mathbb{E}\left[k\left(\mathbf{X}, \mathbf{X}'\right)\right] + \mathbb{E}\left[k\left(\mathbf{Y}, \mathbf{Y}'\right)\right] - 2\mathbb{E}[k(\mathbf{X}, \mathbf{Y})]$$

When the kernel function $k$ degenerates to a unrectified $k_\alpha(x, y) := -\|x - y\|^\alpha$ for $\alpha \in (0, 2)$, the squared MMD would degenerate to

$$\mathbb{E}\|\mathbf{X} - \mathbf{X}'\|^\alpha + \mathbb{E}\|\mathbf{Y} - \mathbf{Y}'\|^\alpha - 2\mathbb{E}\|\mathbf{X} - \mathbf{Y}\|^\alpha$$

On the other hand, we have the Sinkhorn loss as

$$\overline{\mathcal{W}}_{c,\infty}(\mu, \nu) = 2\mathcal{W}_{c,\infty}(\mu, \nu) - \mathcal{W}_{c,\infty}(\nu, \nu) - \mathcal{W}_{c,\infty}(\mu, \nu)$$

Denoting $\Pi_\varepsilon$ be the unique minimizer for $\overline{\mathcal{W}}_{c,\varepsilon}$, it holds that $\Pi_\varepsilon \to \mu \otimes \nu$ as $\varepsilon \to \infty$. That being said, $\mathcal{W}_{c,\infty}(\mu, \nu) \to \int c(x, y)\mathrm{d}\mu(x)\mathrm{d}\nu(y) + 0 = \int c(x, y)\mathrm{d}\mu(x)\mathrm{d}\nu(y)$. If $c = -k_\alpha = \|x - y\|^\alpha$, we eventually have $\mathcal{W}_{-k_\alpha,\infty}(\mu, \nu) \to \int \|x - y\|^\alpha \mathrm{d}\mu(x)\mathrm{d}\nu(y) = \mathbb{E}\|\mathbf{X} - \mathbf{Y}\|^\alpha$. Finally, we can have

$$\overline{\mathcal{W}}_{-k_\alpha,\infty} \to 2\mathbb{E}\|\mathbf{X} - \mathbf{Y}\|^\alpha - \mathbb{E}\|\mathbf{X} - \mathbf{X}'\|^\alpha - \mathbb{E}\|\mathbf{Y} - \mathbf{Y}'\|^\alpha$$

which is exactly the form of squared MMD. Now the key is prove that $\Pi_\varepsilon \to \mu \otimes \nu$ as $\varepsilon \to \infty$.

Firstly, it is apparent that $\mathcal{W}_{c,\varepsilon}(\mu, \nu) \leq \int c(x, y)\mathrm{d}\mu(x)\mathrm{d}\nu(y)$ as $\mu \otimes \nu \in \Pi(\mu, \nu)$. Let $\{\varepsilon_k\}$ be a positive sequence that diverges to $\infty$, and $\Pi_k$ be the corresponding sequence of unique minimizers for $\mathcal{W}_{c,\varepsilon}$. According to the optimality condition, it must be the case that $\int c(x, y)\mathrm{d}\Pi_k + \varepsilon_k \mathrm{KL}(\Pi_k, \mu \otimes \nu) \leq \int c(x, y)\mathrm{d}\mu \otimes \nu + 0$ (when $\Pi(\mu, \nu) = \mu \otimes \nu$). Thus,

$$\mathrm{KL}\left(\Pi_k, \mu \otimes \nu\right) \leqslant \frac{1}{\varepsilon_k}\left(\int c\,\mathrm{d}\mu \otimes \nu - \int c\,\mathrm{d}\Pi_k\right) \to 0.$$

Besides, by the compactness of $\Pi(\mu, \nu)$, we can extract a converging subsequence $\Pi_{n_k} \to \Pi_\infty$. Since KL is weakly lower-semicontinuous, it holds that

$$\mathrm{KL}\left(\Pi_\infty, \mu \otimes \nu\right) \leqslant \lim\inf_{k \to \infty} \mathrm{KL}\left(\Pi_{n_k}, \mu \otimes \nu\right) = 0$$

Hence $\Pi_\infty = \mu \otimes \nu$. That being said that the optimal coupling is simply the product of the marginals, indicating that $\Pi_\varepsilon \to \mu \otimes \nu$ as $\varepsilon \to \infty$. As a special case, when $\alpha = 1$, $\overline{\mathcal{W}}_{-k_1,\infty}(u, v)$ is equivalent to the energy distance

$$d_E(\mathbf{X}, \mathbf{Y}) := 2\mathbb{E}\|\mathbf{X} - \mathbf{Y}\| - \mathbb{E}\|\mathbf{X} - \mathbf{X}'\| - \mathbb{E}\|\mathbf{Y} - \mathbf{Y}'\|. \tag{23}$$

In summary, if the cost function is the rectified kernel $k_\alpha$, it is the case that $\overline{\mathcal{W}}_{-k_\alpha,\varepsilon}$ converges to the squared MMD as $\varepsilon \to \infty$. According to (Nguyen et al., 2020), $\mathfrak{T}^\pi$ is $\gamma^{\alpha/2}$-contractive in the supreme form of MMD with the rectified kernel $k_\alpha$.

For the unspecified $\varepsilon$, we can get the similar result to the case of $\varepsilon \to 0$. For $\forall\delta$, there exists a large positive constant $M$, such that $\mathrm{MMD}^2_{k_\alpha} - \mathcal{W}_{c,\varepsilon} < \delta$ when $\epsilon \geq M$. Based on that, we have the contraction conclusion:

$$\begin{aligned}
\overline{\mathcal{W}}^\infty_{-\kappa_\alpha,\varepsilon}(\mathfrak{T}^\pi Z_1, \mathfrak{T}^\pi Z_2) &= \overline{\mathcal{W}}^\infty_{-\kappa_\alpha,\varepsilon}(\mathfrak{T}^\pi Z_1, \mathfrak{T}^\pi Z_2) - \mathrm{MMD}^2_\infty(\mathfrak{T}^\pi Z_1, \mathfrak{T}^\pi Z_2) + \mathrm{MMD}^2_\infty(\mathfrak{T}^\pi Z_1, \mathfrak{T}^\pi Z_2) \\
&\leq \mathrm{MMD}^2_\infty(\mathfrak{T}^\pi Z_1, \mathfrak{T}^\pi Z_2) - \delta,
\end{aligned} \tag{24}$$

where the first term $\mathrm{MMD}^2_\infty(\mathfrak{T}^\pi Z_1, \mathfrak{T}^\pi Z_2)$ is $\gamma^{\frac{\alpha}{2}}$-contractive, and thus for the unspecified $\varepsilon$, the only difference from the limiting $\varepsilon = \infty$ is an infinitesimal $\delta$, which will vanish as $\varepsilon \to +\infty$ or $(M \to +\infty)$.

**3. For $\varepsilon \in (0, +\infty)$, the contraction property needs a long proof.** The proof pipeline is firstly we prove the three properties of Sinkhorn divergence, and then we show the contraction of distributional Bellman operator under Sinkhorn divergence based on its properties. Most importantly, we analyzed the contraction under a new non-constant factor.

**3.1 Properties of Sinkhorn Divergence.** We recap three crucial properties of a divergence metric. The first is *scale sensitive* **(S)** (of order $\beta$, $\beta > 0$), i.e., $d_p(cX, cY) \leq |c|^\beta d_p(X, Y)$. The second property is *shift invariant* **(I)**, i.e., $d_p(A + X, A + Y) \leq d_p(X, Y)$. The last one is *unbiased gradient* **(U)**. A key observation for the analysis is that the Sinkhorn divergence would degenerate to a two-dimensional KL divergence, and therefore embraces a similar convergence behavior to KL divergence. Concretely, according to the equivalent form of $\mathcal{W}_{c,\varepsilon}(\mu, \nu)$ in Eq. 9, it can be expressed

as the KL divergence between an optimal joint distribution and a Gibbs distribution associated with the cost function:

$$\mathcal{W}_{c,\varepsilon}(\mu,\nu) := \mathrm{KL}\left(\Pi^*(\mu,\nu)|\mathcal{K}(\mu,\nu)\right), \tag{25}$$

where $\Pi^*$ is the optimal joint distribution. Thus, the total Sinkhorn divergence is expressed as

$$\overline{\mathcal{W}}_{c,\varepsilon}(\mu,\nu) := 2\mathrm{KL}\left(\Pi^*(\mu,\nu)|\mathcal{K}(\mu,\nu)\right) - \mathrm{KL}\left(\Pi^*(\mu,\mu)|\mathcal{K}(\mu,\mu)\right) - \mathrm{KL}\left(\Pi^*(\nu,\nu)|\mathcal{K}(\nu,\nu)\right). \tag{26}$$

Due to the form of $\overline{\mathcal{W}}_{c,\varepsilon}(\mu,\nu)$, the convergence behavior is determined by $\mathcal{W}_{c,\varepsilon}(\mu,\nu)$, which is similar to the behavior of KL divergence. Thus, we will focus on the convergence analysis of $\mathcal{W}_{c,\varepsilon}(\mu,\nu)$. According to the fact that KL divergence has unbiased gradient estimates (**U**) and shift invariant (**I**), and Sinkkhorn divergence can be viewed as a two-dimensional KL divergence, both properties of **U** and *I* can be extended to Sinkhorn divergence. However, we find the non scale sensitive (**S**) property can not directly apply to Sinkhorn divergence due to the minimum nature of $\mathcal{W}_{c,\varepsilon}(\mu,\nu)$ as the optimal joint distribution $\Pi^*(\mu,\nu)$ could be different from $\Pi^0(a\mu,a\nu)$ where $a$ is the scale factor. We need a new rigorous proof of scale sensitive property as follows.

### 3.2 Scale Sensitive Property of Sinkhorn Divergence.

We show Sinkhorn divergence satisfies a variant of scale sensitive property when $c = -k_\alpha$ that corresponds to a non-constant scale factor $\Delta(a,\alpha)$ that is not only a function of the vanilla scale factor $a$ and $\alpha$ in $k_\alpha$, but also the two specified probability measures $(U,V)$. By definition, the pdf of $\mathcal{K}(U,V) \propto e^{\frac{-c(x,y)}{\varepsilon}}\mu(x)\nu(y)$. After a scaling transformation, the pdf of $aU$ and $aV$ with respect to $x$ and $y$ would be $\frac{1}{a}\mu(\frac{x}{a})$ and $\frac{1}{a}\nu(\frac{y}{a})$. Thus $\mathcal{K}(aU,aV) \propto e^{\frac{-c(x,y)}{\varepsilon}}\frac{1}{a}\mu(\frac{x}{a})\frac{1}{a}\nu(\frac{y}{a})$. We denote $\Pi^*$ and $\Pi^0$ as the optimal joint distribution of $\mathcal{W}_{c,\varepsilon}(\mu,\nu)$ and $\mathcal{W}_{c,\varepsilon}(a\mu,a\nu)$.

$$
\begin{aligned}
\mathcal{W}_{c,\varepsilon}(aU,aV) &= \int c(x,y)\mathrm{d}\Pi^0(x,y) + \varepsilon\mathrm{KL}(\Pi^0|a\mu\otimes a\nu) \\[2mm]
&\leq \int c(x,y)\mathrm{d}\Pi^*(x,y) + \varepsilon\mathrm{KL}(\Pi^*|a\mu\otimes a\nu) \\[2mm]
&\overset{c=-k_\alpha}{=} \int (x-y)^\alpha \frac{1}{a^2}\pi^*(\frac{x}{a},\frac{y}{a})\mathrm{d}x\mathrm{d}y + \varepsilon\int \frac{1}{a^2}\pi^*(\frac{x}{a},\frac{y}{a})\log\frac{\frac{1}{a^2}\pi^*(\frac{x}{a},\frac{y}{a})}{\frac{1}{a^2}\mu(\frac{x}{a})\nu(\frac{y}{a})}\mathrm{d}x\mathrm{d}y \\[2mm]
&= |a|^\alpha \int (x-y)^\alpha \pi^*(x,y)\mathrm{d}x\mathrm{d}y + \varepsilon\int \pi^*(x,y)\log\frac{\pi^*(x,y)}{\mu(x)\nu(y)}\mathrm{d}x\mathrm{d}y \\[2mm]
&= \int (x-y)^\alpha \pi^*(x,y)\mathrm{d}x\mathrm{d}y + \varepsilon\mathrm{KL}(\Pi^*|\mu\otimes\nu) - (1-|a|^\alpha)\int (x-y)^\alpha \pi^*(x,y)\mathrm{d}x\mathrm{d}y \\[2mm]
&= \mathcal{W}_{c,\varepsilon}(U,V) - (1-|a|^\alpha)\int (x-y)^\alpha \mathrm{d}\Pi^*(x,y) \\[2mm]
&= \Delta^{U,V}(a,\alpha)\mathcal{W}_{c,\varepsilon}(U,V)
\end{aligned}
\tag{27}
$$

where $\Delta^{U,V}(a,\alpha) = 1 - \frac{(1-|a|^\alpha)\int (x-y)^\alpha \mathrm{d}\Pi^*(x,y)}{\mathcal{W}_{c,\varepsilon}(U,V)} \in (0,1)$ for $\varepsilon \in (0,+\infty)$ and $a < 1$ due to the fact that $0 < (1-|a|^\alpha)\int (x-y)^\alpha \mathrm{d}\Pi^*(x,y) < \int (x-y)^\alpha \mathrm{d}\Pi^*(x,y) < \mathcal{W}_{c,\varepsilon}(U,V)$. $\Delta^{U,V}(a,\alpha)$ is function less than 1 that depends on the two margin distributions and the scale factor $a$. The result implies that we have a new variant of scale sensitive property of Sinkhorn divergence with a non-constant factor $\Delta^{U,V}(a,\alpha) < 1$ when we choose $c = -k_\alpha$ and $|a| < 1$.

### 3.3 A New Contraction Mapping Theorem.

We derive a new contraction mapping theorem based on the distribution distance $d$ in order to prove the convergence in 3.4.

**Theorem 2.** *(Distribution Contraction Mapping Theorem with a Non-constant Factor) Consider a distribution distance $d$ and a function $g : \mathcal{P} \to \mathcal{P}$. The mapping $d$ is a contraction: There exists a function $q(X,Y) < 1$ such that for $\forall$ distributions $X$ and $Y$:*

$$d(g(X),g(Y)) \leq q(X,Y)d(X,Y) \tag{28}$$

*Then there exists a unique distribution $X^*$ with $g(X^*) = X^*$.*

*Proof.* We consider the convergence of the distribution sequence $X^k$. We have the updating rule as

$$d(X^{k+1}, X^k) = d(g(X^k), g(X^{k-1})) \leq q_{k,k-1} d(X^k, X^{k-1}), \tag{29}$$

where we use $q_{k,k-1} = q(X^k, X^{k-1})$ for short. Hence, we have

$$d(X^{k+1}, X^k) \leq \Pi_{i=1}^k q_{i,i-1} d(X^1, X^0). \tag{30}$$

Let $d_0 = d(X^1, X^0)$. From the triangle inequality, we have

$$
\begin{aligned}
d(X^{k+l}, X^k) &\leq d(X^{k+1}, X^k) + ... + d(X^{k+l}, X^{k+l-1}), \\
&\leq \Pi_{i=1}^k q_{i,i-1} d_0 + .. + \Pi_{i=1}^{k+l-1} q_{i,i-1} d_0 \\
&\leq \Pi_{i=1}^k q_{i,i-1} (1 + q_{k+1,k} + ... + \Pi_{i=k+1}^{k+l-1} q_{i,i-1}) d_0 \\
&\leq \Pi_{i=1}^k q_{i,i-1} (1 + q_{k+1,k} + ... + \Pi_{i=k+1}^{k+l-1} q_{i,i-1} + ...) d_0
\end{aligned}
\tag{31}
$$

For the infinite series $1 + q_{k+1,k} + ... + \Pi_{i=k+1}^{k+l-1} q_{i,i-1} + ...$, which we denote as $u_i$ for $i$-the term, according to the ratio convergence judgment method of infinite series, $\lim_{k\to\infty} \frac{u_{i+1}}{u_i} < 1$. Thus, the infinite series is convergent. Due to the fact $\Pi_{i=1}^k q_{i,i-1} \to 0$ as $k \to \infty$, we have $d(X^{k+1}, X^k) \to 0$ as $k \to \infty$. Therefore, it must converge to a limit distribution $X^*$ that satisfies $g(X^*) = X^*$. $\square$

### 3.4 Contraction of Distributional Bellman Operator under Sinkhorn Divergence.

According to the equation of $\overline{\mathcal{W}}_{c,\varepsilon}$, it holds the same properties as $\mathcal{W}_{c,\varepsilon}$, i.e., shift invariant and scale sensitive. Thus, we derive the convergence of distributional Bellman operator $\mathfrak{T}^\pi$ under the supreme form of $\overline{\mathcal{W}}_{c,\varepsilon}$, i.e., $\overline{\mathcal{W}}_{c,\varepsilon}^\infty$:

$$
\begin{aligned}
&\overline{\mathcal{W}}_{c,\varepsilon}^\infty(\mathfrak{T}^\pi Z_1, \mathfrak{T}^\pi Z_2) \\
&= \sup_{s,a} \overline{\mathcal{W}}_{c,\varepsilon}(\mathfrak{T}^\pi Z_1(s,a), \mathfrak{T}^\pi Z_2(s,a)) \\
&= \overline{\mathcal{W}}_{c,\varepsilon}(R(s,a) + \gamma Z_1(s',a'), R(s,a) + \gamma Z_2(s',a')) \\
&\overset{c=-k_\alpha}{\leq} \Delta^{Z_1(s',a'),Z_2(s',a')}(\gamma, \alpha) \overline{\mathcal{W}}_{c,\varepsilon}(Z_1(s',a'), Z_2(s',a')) \\
&\leq \sup_{s',a'} \Delta^{Z_1(s',a'),Z_2(s',a')}(\gamma, \alpha) \sup_{s',a'} \overline{\mathcal{W}}_{c,\varepsilon}(Z_1(s',a'), Z_2(s',a')) \\
&\leq \Delta^{Z_1,Z_2}(\gamma, \alpha) \sup_{s',a'} \overline{\mathcal{W}}_{-k_\alpha,\varepsilon}(Z_1(s',a'), Z_2(s',a')) \\
&= \Delta^{Z_1,Z_2}(\gamma, \alpha) \overline{\mathcal{W}}_{-k_\alpha,\varepsilon}^\infty(Z_1, Z_2)
\end{aligned}
\tag{32}
$$

where the first inequality comes from the scale sensitive property proof of Sinkhorn divergence and we let $\Delta^{Z_1,Z_2}(\gamma, \alpha) = \sup_{s',a'} \Delta^{Z_1(s',a'),Z_2(s',a')}(\gamma)$. If $\Delta^{Z_1,Z_2}(\gamma, \alpha)$ is only a constant function in terms of $\gamma$ and $\alpha$, we can directly arrive the conclusion that distributional Bellman operator is $\Delta^{Z_1,Z_2}(\gamma, \alpha)$-contractive based on existing Banach fixed point theorem. However, the fact is that $\Delta^{Z_1,Z_2}(\gamma, \alpha)$ also depends on $Z_1$ and $Z_2$, and thus we need a new contraction mapping theorem to guarantee the convergence of fixed distribution iteration. According to Theorem 2 in 3.3 that we specifically figure out for the our contraction proof, we have $\overline{\mathcal{W}}_{c,\varepsilon}^\infty$ can guarantee the convergence via distributional Bellman iterations. In summary, we conclude that $\mathfrak{T}^\pi$ is a contractive operator when we use the $-k_\alpha$ as the cost function and $\gamma \leq 1$, **while the contraction factor, which is short for $\Delta(\gamma, \alpha) < 1$, is not only a function of $\alpha$ and $\gamma$, but also depends on distribution sequence in the while iterations.**

$\square$

# D   PROOF OF PROPOSITION 1 AND COROLLARY 1

*Proof.* As we leverage $\Pi^*$ to denote the optimal $\Pi$ by evaluating the Sinkhorn divergence via $\min_{\Pi \in \Pi(\mu,\nu)} \overline{\mathcal{W}}_{c,\varepsilon}(\mu,\nu;k)$, the Sinkhorn divergence can be composed in the following form:

$$
\begin{aligned}
&\overline{\mathcal{W}}_{c,\varepsilon}(\mu,\nu;k) \\
&= 2\mathrm{KL}\left(\Pi^*(\mu,\nu)|\mathcal{K}_{-k}(\mu,\nu)\right) - \mathrm{KL}\left(\Pi^*(\mu,\mu)|\mathcal{K}_{-k}(\mu,\mu)\right) - \mathrm{KL}\left(\Pi^*(\nu,\nu)|\mathcal{K}_{-k}(\nu,\nu)\right) \\
&= 2(\mathbb{E}_{X,Y}\left[\log \Pi^*(\mu,\nu)\right] + \frac{1}{\varepsilon}\mathbb{E}_{X,X'}\left[c(X,Y)\right]) - (\mathbb{E}_{X,X'}\left[\log \Pi^*(\mu,\nu)\right] + \frac{1}{\varepsilon}\mathbb{E}_{X,Y}\left[c(X,Y)\right]) \\
&\quad - (\mathbb{E}_{Y,Y'}\left[\log \Pi^*(\nu,\nu)\right] + \frac{1}{\varepsilon}\mathbb{E}_{Y,Y'}\left[c(Y,Y')\right]) \\
&= \mathbb{E}_{X,X',Y,Y'}\left[\log \frac{(\Pi^*(X,Y))^2}{\Pi^*(X,X')\Pi^*(Y,Y')}\right] + \frac{1}{\varepsilon}(\mathbb{E}_{X,X'}\left[k(X,X')\right] + \mathbb{E}_{Y,Y'}\left[k(Y,Y')\right] - 2\mathbb{E}_{X,X'}\left[k(X,Y)\right]) \\
&= \mathbb{E}_{X,X',Y,Y'}\left[\log \frac{(\Pi^*(X,Y))^2}{\Pi^*(X,X')\Pi^*(Y,Y')}\right] + \frac{1}{\varepsilon}\mathrm{MMD}^2_{-c}(\mu,\nu)
\end{aligned}
\tag{33}
$$

where the cost function $c$ in the Gibbs distribution $\mathcal{K}$ is minus Gaussian kernel, i.e., $c(x,y) = -k(x,y) = e^{-(x-y)/(2\sigma^2)}$. Till now, we have shown the result in Corollary 1.

Next, we use Taylor expansion to prove the moment matching of MMD. Firstly, we have the following equation:

$$
\begin{aligned}
\mathrm{MMD}^2_{-c}(\mu,\nu) &= \mathbb{E}_{X,X'}\left[k(X,X')\right] + \mathbb{E}_{Y,Y'}\left[k(Y,Y')\right] - 2\mathbb{E}_{X,X'}\left[k(X,Y)\right] \\
&= \mathbb{E}_{X,X'}\left[\phi(X)^\top \phi(X')\right] + \mathbb{E}_{Y,Y'}\left[\phi(Y)^\top \phi(Y')\right] - 2\mathbb{E}_{X,X'}\left[\phi(X)^\top \phi(Y)\right] \\
&= \mathbb{E}\|\phi(X) - \phi(Y)\|^2
\end{aligned}
\tag{34}
$$

We expand the Gaussian kernel via Taylor expansion, i.e.,

$$
\begin{aligned}
k(x,y) &= e^{-(x-y)^2/(2\sigma^2)} \\
&= e^{-\frac{x^2}{2\sigma^2}} e^{-\frac{y^2}{2\sigma^2}} e^{\frac{xy}{\sigma^2}} \\
&= e^{-\frac{x^2}{2\sigma^2}} e^{-\frac{y^2}{2\sigma^2}} \sum_{n=0}^{\infty} \frac{1}{\sqrt{n!}}(\frac{x}{\sigma})^n \frac{1}{\sqrt{n!}}(\frac{y}{\sigma})^n \\
&= \sum_{n=0}^{\infty} e^{-\frac{x^2}{2\sigma^2}} \frac{1}{\sqrt{n!}}(\frac{x}{\sigma})^n e^{-\frac{y^2}{2\sigma^2}} \frac{1}{\sqrt{n!}}(\frac{y}{\sigma})^n \\
&= \phi(x)^\top \phi(y)
\end{aligned}
\tag{35}
$$

Therefore, we have

$$
\begin{aligned}
\mathrm{MMD}^2_{-c}(\mu,\nu) &= \sum_{n=0}^{\infty} \frac{1}{\sigma^{2n}n!}\left(\mathbb{E}_{x\sim\mu}\left[e^{-x^2/(2\sigma^2)}x^n\right] - \mathbb{E}_{x\sim\nu}\left[e^{-y^2/(2\sigma^2)}y^n\right]\right)^2 \\
&= \sum_{n=0}^{\infty} \frac{1}{\sigma^{2n}n!}\left(\tilde{M}_n(\mu) - \tilde{M}_n(\nu)\right)^2
\end{aligned}
\tag{36}
$$

$\tilde{M}_n(\mu) = \mathbb{E}_{x\sim\mu}\left[e^{-x^2/(2\sigma^2)}x^n\right]$, and similarly for $\tilde{M}_n(\nu)$. The conclusion is the same as the moment matching in (Nguyen et al., 2020). Finally, due to the equivalence of $\overline{\mathcal{W}}_{c,\varepsilon}(\mu,\nu)$ after multiplying $\varepsilon$, we have

$$
\begin{aligned}
\overline{\mathcal{W}}_{c,\varepsilon}(\mu,\nu;k) &:= \mathrm{MMD}^2_{-c}(\mu,\nu) + \varepsilon\mathbb{E}\left[\frac{(\Pi^*(X,Y))^2}{\Pi^*(X,X')\Pi^*(Y,Y')}\right] \\
&= \sum_{n=0}^{\infty} \frac{1}{\sigma^{2n}n!}\left(\tilde{M}_n(\mu) - \tilde{M}_n(\nu)\right)^2 + \varepsilon\mathbb{E}\left[\frac{(\Pi^*(X,Y))^2}{\Pi^*(X,X')\Pi^*(Y,Y')}\right],
\end{aligned}
\tag{37}
$$

This result is also equivalent to Theorem 1, where $\Pi^*$ would degenerate to $\mu \otimes \nu$ as $\varepsilon \to +\infty$. In that case, the first regularization term would vanish, and thus the Sinkhorn divergence degrades to a MMD loss, i.e., $\mathrm{MMD}^2_{-c}(\mu, \nu)$.

$\square$

## E    HUMAN-NORMALIZED SCORES

|  | **Mean** | **Median** | $>$ **Human** | $>$**DQN** |
|---|---|---|---|---|
| DQN | 438.7 % | 43.6 % | 17 | 0 |
| C51 | 1043.4 % | 103.7 % | 26 | 42 |
| QR-DQN-1 | 1286.4 % | 108.6 % | 31 | 47 |
| MMDDRL | 924.6 % | **117.5** % | 27 | 43 |
| QRDQN(tf) | 535.1 % | 108.2 % | 28 | 40 |
| MMDDRL(tf) | 665.0 % | 99.8 % | 27 | 39 |
| SinkhornDRL | **1435.8 %** | 113.0 % | 27 | 42 |

Table 2: Mean and median of best human-normalized scores across 55 Atari 2600 games. The results for all considered algorithms are averaged over 3 seeds. DQN, C51, QR-DQN-1, MMDDRL are based on our **Pytorch implementation** after 10M time steps adapted from (Zhang, 2018), while QRDQN(tf) and MMDDRL(tf) are training results also after 10M time steps from the **tensorflow Dopamine framework** of MMDDRL (Nguyen et al., 2020) released in (Nguyen-Tang, 2021).

Human normalized score equation is (algorithm - randomplay) / (human - randomplay). Our implementation of DQN, QRDQN-1, C51, MMDDRL, Sinkhorn is based on (Zhang, 2018) and all the experimental settings, including parameters are identical to the distributional RL baselines implemented by (Zhang, 2018). The results about mean and median human-normalized scores of all considered distributional RL algorithms are reported in Table 2. We also compare the performance of QRDQN(tf) and MMDDRL(tf) after the same 10M time steps based on tensorflow implementation on Dopamine framework (Castro et al., 2018). These results are averaged over training data released in (Nguyen-Tang, 2021). We argue that Human-normalized scores may be limited to evaluate the superiority of algorithms as mean can be highly affected by the performance on games with high-level returns. For example, in Figure 2, MMDDRL is superior to QR-DQN as Sinkhorn outperforms MMDDRL on a smaller portion of games compared with QRDQN, but the mean score for MMDDRL in Table 2 is lower than QRDQN. By contrast, SinkhornDRL is superior in terms of mean score and competitive in terms of median. **We also provide all average results in Table 3 of Appendix H and all learning curves of our implemented algorithms in Appendix F.**

## F    MORE EXPERIMENTAL RESULTS

We provide learning curves of DQN, QRDQN, C51, MMD and SinkhornDRL algorithms on all 55 Atari games in Figures 4 5 6 7 8 9. It illustrates that SinkhornDRL dramatically surpasses the other distributional RL algorithms on a large amount of environments, e.g., Venture, Atlantis, Tennis and SpaceInvader, and presents competitive performance or is only slightly inferior as opposed to the state-of-the-art baselines on other games. Note that the average improvement of SinkhornDRL on Venture game is significant owing to one to two times convergence of SinkhornDRL algorithm over 3 seeds, while the other baselines do not converge over the considered seeds. Although this improvement may also suffer from the instability issue, its occasional success for our SinkhornDRL algorithm also presents huge potential on some complicated environments. We leave the further exploration on the advantage and potential of SinkhornDRL algorithm as the future work.

## G    SENSITIVITY ANALYSIS AND COMPUTATIONAL COST

### G.1    MORE RESULTS IN SENSITIVITY ANALYSIS

**Decreasing** $\varepsilon$**.**    We argue that the limit behavior connection as stated in Theorem 1 may not be able to be verified rigorously via numeral experiments due to the numerical instability of Sinkhorn

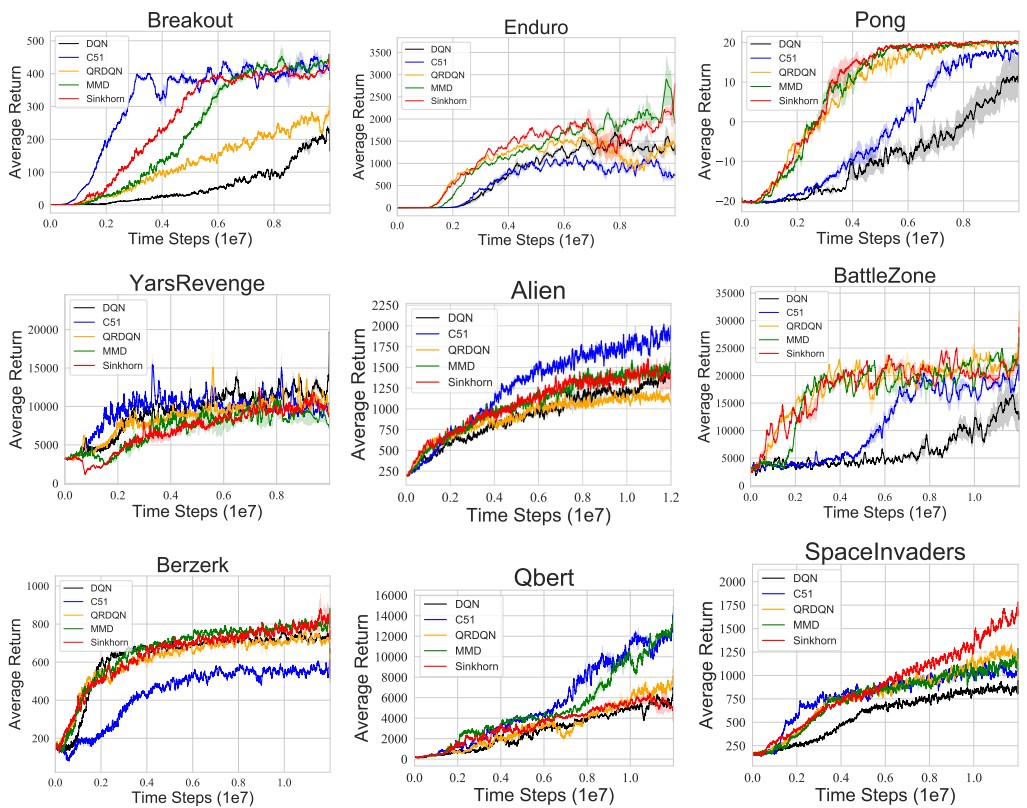

Figure 4: Performance of SinkhornDRL compared with DQN, C51, QRDQN and MMD on Breakout, Enduro, Pong, YarRevenge, Alien, BattleZone, Berzerk, Qbert and SpaceInvader.

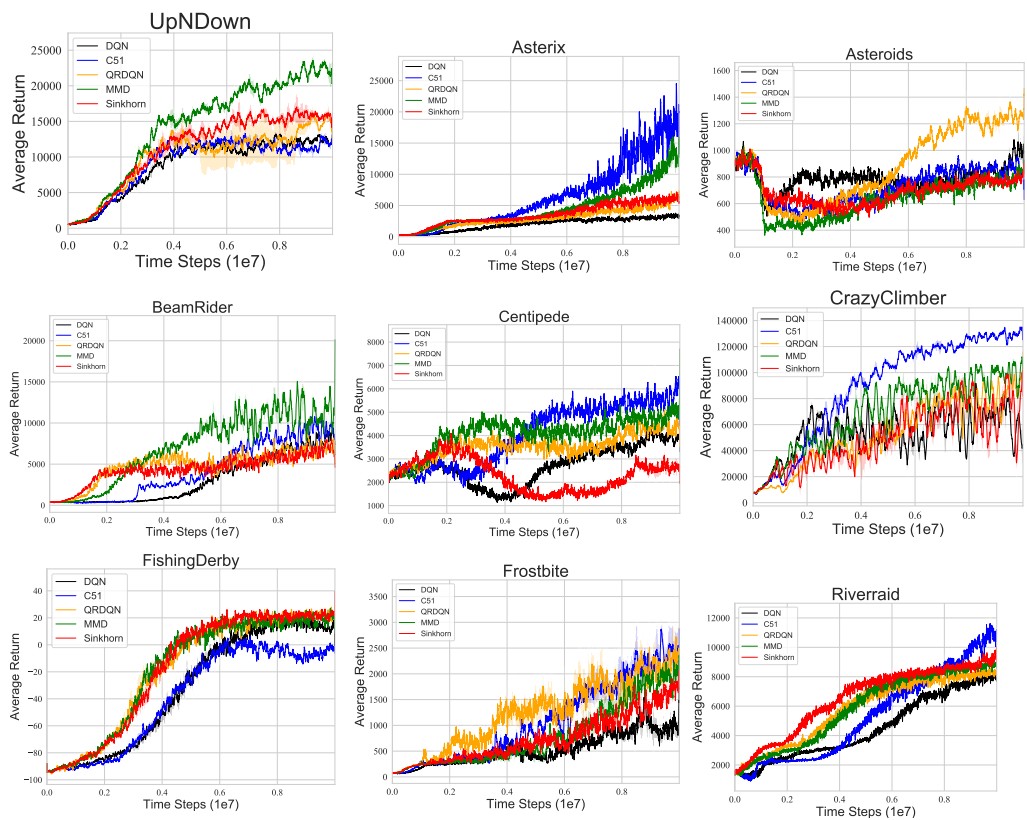

Figure 5: Performance of SinkhornDRL compared with DQN, C51, QRDQN and MMD on UpN-Down, Asterix, Asteriods, BeamRider, Centipede, FishingDerby, Frostbite and Riverraid.

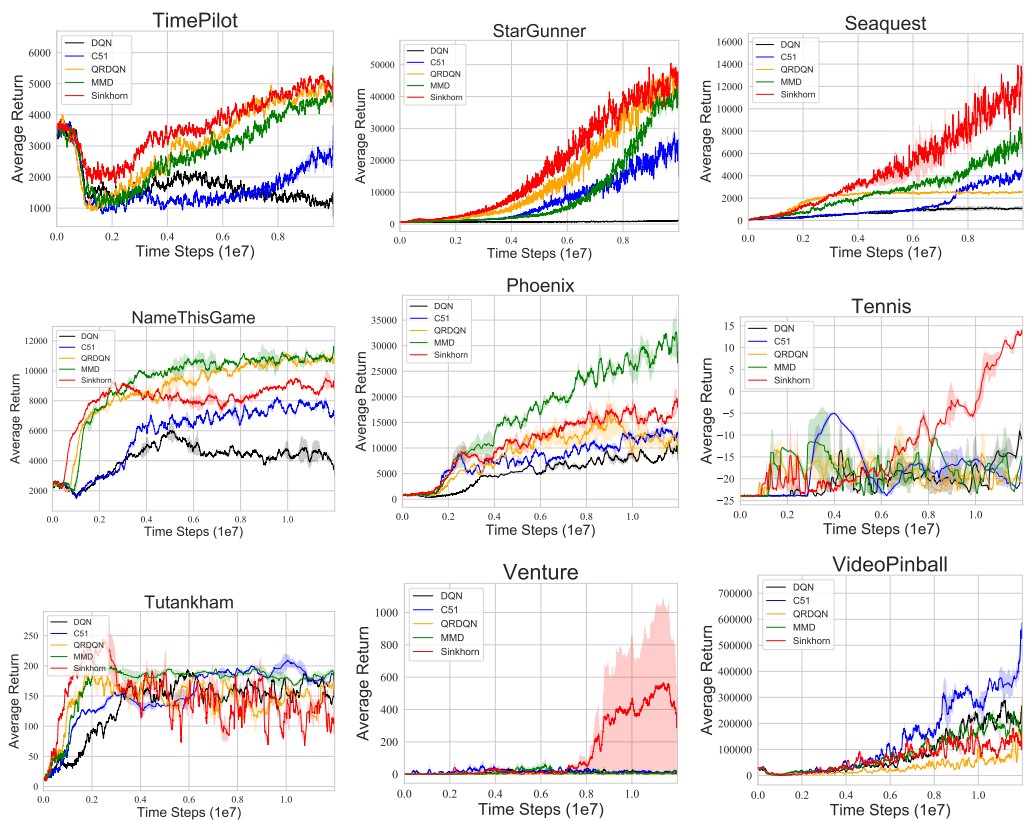

Figure 6: Performance of SinkhornDRL compared with DQN, C51, QRDQN and MMD on TimePilot, StarGuner, Seaquest, NameThisGame, Phoenix, Tennix, Tutankham, Venture and VideoPinball.

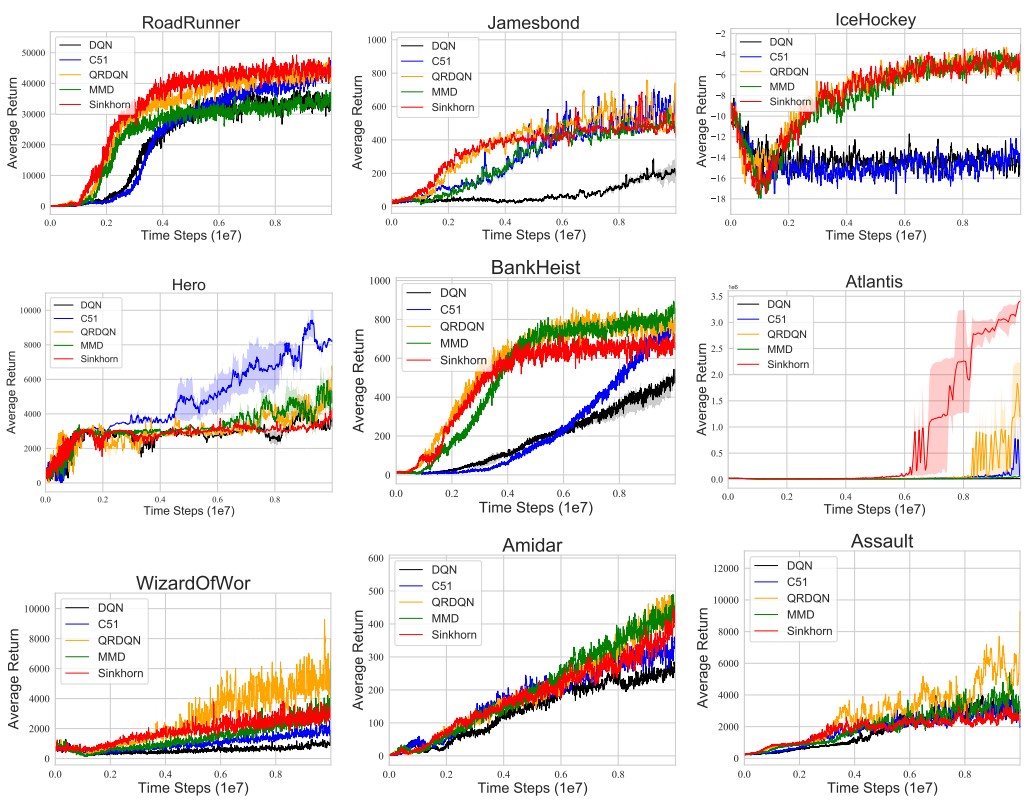

Figure 7: Performance of SinkhornDRL compared with DQN, C51, QRDQN and MMD on Road-Runner, Jamesbond, IceHockey, Hero, BankHeist, Atlantis, WizardOfWor, Amidar and Assault.

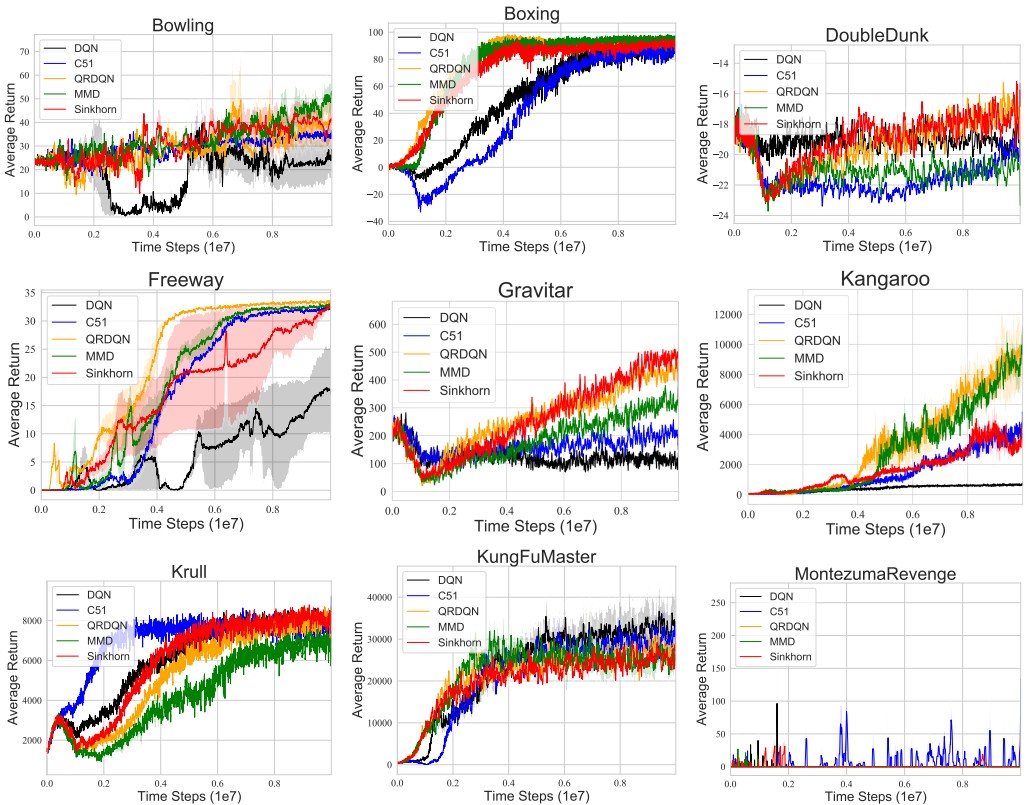

Figure 8: Performance of SinkhornDRL compared with DQN, C51, QRDQN and MMD on Bowling, Boxing, DoubleDunk, Freeway, Gravitar, Kangaroo, Krull, KunFuMaster and MontezumaRevenge.

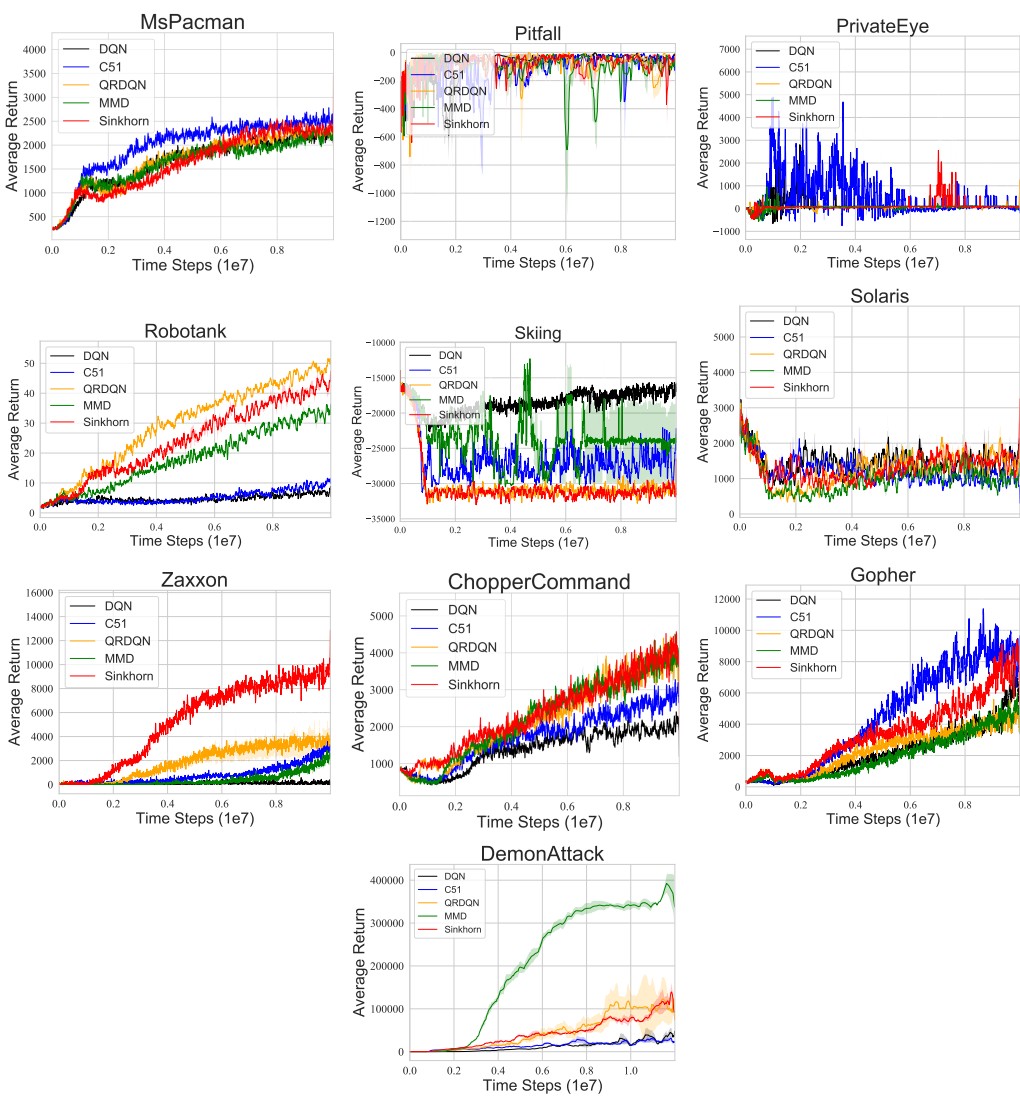

Figure 9: Performance of SinkhornDRL compared with DQN, C51, QRDQN and MMD on MsPac-man, Pitfall, PrivateEye, Robotank, Skiing, Solaris, Zaxxon, ChopperCommand, Gopher and De-monAttack.

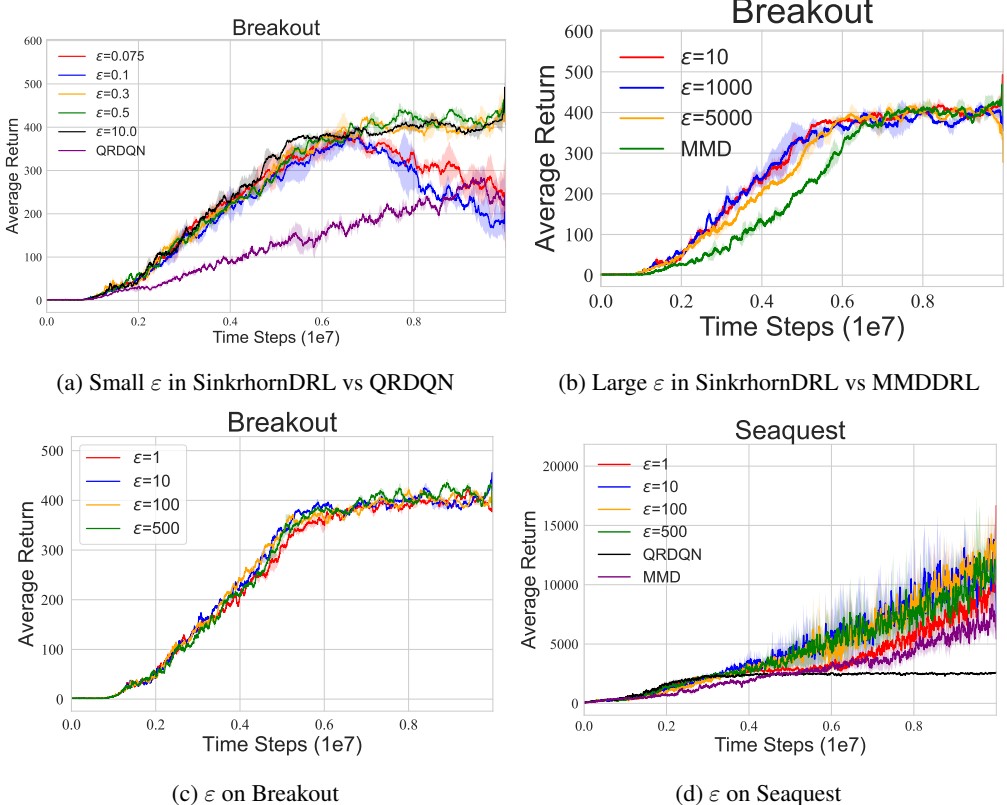

Figure 10: (a) Sensitivity analysis w.r.t. a small level of $\varepsilon$ SinkhornDRL to compare with QR-DQN that approximates Wasserstein distance on Breakout. (b) Sensitivity analysis w.r.t. a large level of $\varepsilon$ SinkhornDRL algorithm to compare with MMDDRL on Breakout. All learning curves are reported over 2 seeds. (c) and (d) are results for a general $\varepsilon$ on Breakout and Seaquest, respectively.

Iteration in Algorithm 2. From Figure 10 (a), we can observe that if we gradually decline $\varepsilon$ to 0, SinkhornDRL's performance tends to degrade and approach to QR-DQN. Note that an overly small $\varepsilon$ will lead to a trivial almost 0 $\mathcal{K}_{i,j}$ in Sinkhorn iteration in Algorithm 2, and will cause $\frac{1}{0}$ numerical instability issue for $a_l$ and $b_l$ in Line 5 of Algorithm 2. In addition, we also conducted experiments on Seaquest, the similar result is also observed in Figure 10 (d). As shown in Figure 10 (d), the performance of SinkhornDRL is robust when $\varepsilon = 10, 100, 500$, but a small $\epsilon = 1$ tends to worsen the performance.

**Increasing $\varepsilon$.** Moreover, for breakout, if we increase $\varepsilon$, the performance of SinkhornDRL tends to degrade and be close to MMDDRL as suggested in Figure 10 (b). It is also noted that an overly large $\varepsilon$ will let the $\mathcal{K}_{i,j}$ explode to $\infty$. This also leads to numerical instability issue in Sinkhorn iteration in Algorithm 2.

**Samples $N$.** We find that SinkhornDRL requires a proper number of samples $N$ to perform favorably, and the sensitivity w.r.t $N$ depends on the environment. As suggested in Figure 11 (a), a smaller $N$, e.g., $N = 2$ on breakout has already achieved favorably performance and even accelerates the convergence in the early phase, while $N = 2$ on Seaquest will lead to the divergence issue. Meanwhile, an overly large $N$ worsens the performance across two games. We conjecture that using larger network networks to generate more samples may suffer from the overfitting issue, yielding the training instability (Bjorck et al., 2021). In practice, we choose a proper number of sample, i.e., $N = 200$ across all games.

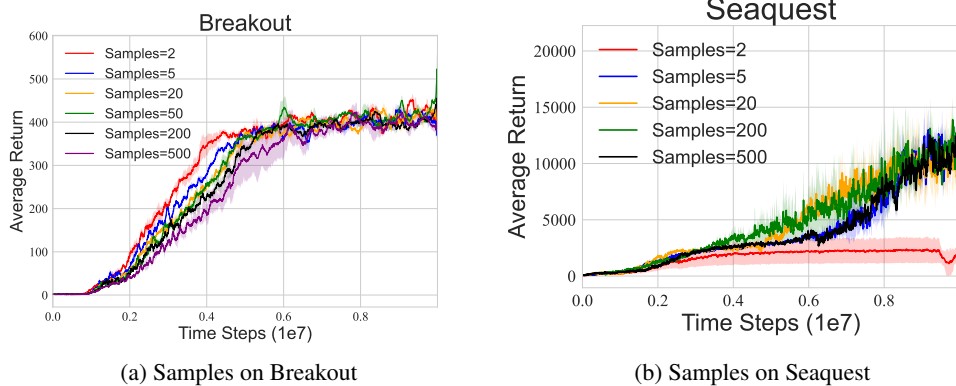

(a) Samples on Breakout

(b) Samples on Seaquest

Figure 11: Sensitivity analysis of Sinkhorn in terms of the number of samples $N$ on Breakout (a) and Seaquest (b).

### G.2 COMPARISON WITH THE COMPUTATIONAL COST

We evaluate the computational time every 10,000 iterations across the whole training process of all considered distributional RL algorithms and make a comparison in Figure 12. It suggests that SinkhornDRL indeed increases around 50% computation cost compared with QR-DQN and C51, but only slightly increases the the cost in contrast to MMDDRL on both Breakout and Qbert games. We argue that this additional computational burden can be tolerant in view of the significant outperformance of SinkhornDRL in a large amount of environments.

In addition, we also find that the number of Sinkhorn iterations $L$ is negligible to the computation cost, while an overly large samples $N$, e.g., 500, will lead to a large computational burden as illustrated in Figure 13. This can be intuitively explained as the computation complexity of the cost function $c_{i,j}$ is $\mathcal{O}(N^2)$ in SinkhornDRL, which is particularly heavy in computation if $N$ is large enough.

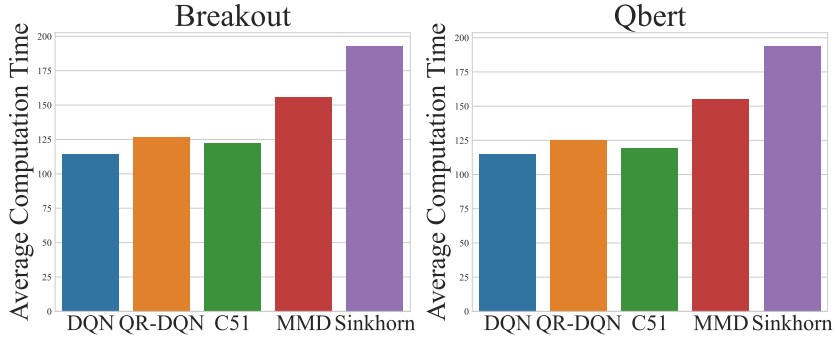

Figure 12: Average computational cost per 10,000 iterations of all considered distributional RL algorithm, where we select $\varepsilon = 10$, $L = 10$ and number of samples $N = 200$ in SinkhornDRL algorithm.

## H   RAW SCORE TABLES ACROSS ALL ATARI GAMES AFTER 10 TIMESTEPS (40M FRAMES)

For distributional Rl algorithms, results are averaged over 3 seeds after 10 timeSteps, i.e., 40M Frames. Results of DQN, C51, QRDQN, MMD and Sinkhorn are based on our implementation on Pytorch, which is adapted from the reliable open-source implementation (Zhang, 2018). QRDQN(tf) and MMD(tf) are from training results after 40M frames from the implementation of MMDDRL (Nguyen et al., 2020) in (Nguyen-Tang, 2021).

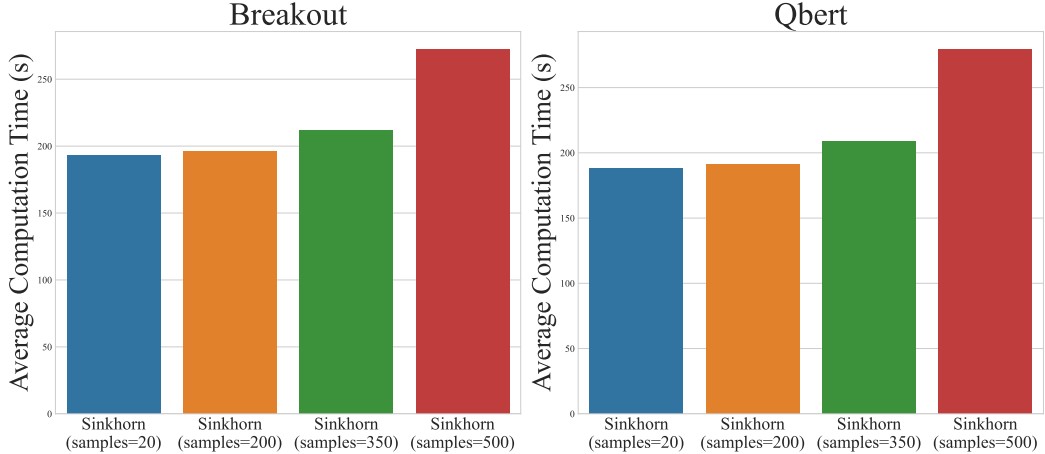

Figure 13: Average computational cost per 10,000 iterations of SinkhornDRL algorithm over different samples.

| GAMES | RANDOM | HUMAN | DQN | C51 | QRDQN | QRDQN(tf) | MMD | MMD(tf) | Sinkhorn |
|---|---|---|---|---|---|---|---|---|---|
| Alien | 211.9 | 7,127.7 | 1334.0 | 1946.0 | 1625.0 | 1425.8 | 2218.0 | 1683.1 | 1873.0 |
| Amidar | 2.34 | 1,719.5 | 400.2 | 354.5 | 554.6 | 870.2 | 706.4 | 694.7 | 506.7 |
| Assault | 283.5 | 742.0 | 5651.8 | 3368.1 | 7593.6 | 7931.8 | 6001.5 | 8947.5 | 3771.0 |
| Asterix | 268.5 | 8,503.3 | 5490.0 | 31860.0 | 7660.0 | 17126.8 | 15890.0 | 36411.5 | 7610.0 |
| Asteroids | 1008.6 | 47,388.7 | 1246.0 | 826.0 | 1660.0 | 1610.7 | 1095.0 | 1460.9 | 624.0 |
| Atlantis | 22188 | 29,028.1 | 18990.0 | 1490040.0 | 2520080.0 | 859419.7 | 80920.0 | 935273.3 | 3417430.0 |
| BankHeist | 14 | 753.1 | 657.0 | 948.0 | 1000.0 | 936.7 | 1034.0 | 982.8 | 849.0 |
| BattleZone | 3000 | 37,187.5 | 22100.0 | 28400.0 | 37800.0 | 25223.2 | 28400.0 | 20089.1 | 27000.0 |
| BeamRider | 414.3 | 16,926.5 | 9519.0 | 13069.2 | 8043.8 | 9728.9 | 14072.6 | 16889.1 | 9865.6 |
| Berzerk | 165.6 | 2,630.4 | 746.0 | 824.0 | 928.0 | 766.3 | 959.0 | 802.3 | 1029.0 |
| Bowling | 23.48 | 160.7 | 29.6 | 30.3 | 35.5 | 33.8 | 60.0 | 37.3 | 12.6 |
| Boxing | -0.69 | 12.1 | 96.0 | 91.8 | 98.3 | 97.9 | 96.9 | 92.3 | 96.7 |
| Breakout | 1.5 | 30.5 | 313.4 | 373.0 | 361.4 | 391.0 | 405.9 | 486.9 | 402.5 |
| Centipede | 2064.77 | 12,017.0 | 4548.1 | 6090.9 | 5508.0 | 5866.0 | 5152.0 | 5885.6 | 4952.2 |
| ChopperCommand | 794 | 7,387.8 | 2780.0 | 4360.0 | 5490.0 | 3575.7 | 6760.0 | 2465.8 | 6520.0 |
| CrazyClimber | 8043 | 35,829.4 | 15960.0 | 158070.0 | 69430.0 | 85543.6 | 112130.0 | 115997.2 | 16000.0 |
| DemonAttack | 162.25 | 1,971.0 | 58324.5 | 41656.5 | 63889.0 | 94433.0 | 437760.5 | 98849.8 | 195827.0 |
| DoubleDunk | -18.14 | -16.4 | 0.2 | 0.6 | -0.4 | -15.1 | -0.4 | -14.9 | -2.2 |
| Enduro | 0.01 | 860.5 | 1961.3 | 1507.5 | 2832.5 | 1912.4 | 3248.2 | 1839.5 | 4272.0 |
| FishingDerby | -93.06 | -38.7 | 15.8 | 26.0 | 33.4 | 18.9 | 24.5 | 21.6 | 24.6 |
| Freeway | 0.01 | 29.6 | 30.9 | 32.6 | 34.0 | 33.1 | 33.6 | 33.2 | 34.0 |
| Frostbite | 73.2 | 4,334.7 | 1767.0 | 3317.0 | 4487.0 | 3359.6 | 2874.0 | 3671.9 | 2632.0 |
| Gopher | 364 | 2,412.5 | 7058.0 | 9314.0 | 6466.0 | 3854.7 | 6412.0 | 4966.6 | 15168.0 |
| Gravitar | 226.5 | 3,351.4 | 110.0 | 325.0 | 565.0 | 509.5 | 345.0 | 547.7 | 470.0 |
| Hero | 551 | 30,826.4 | 4657.5 | 8098.0 | 11673.5 | 9779.7 | 7215.0 | 8382.4 | 7476.0 |
| IceHockey | -10.3 | 0.9 | -13.0 | -11.4 | -3.6 | -3.7 | -4.5 | -2.4 | -4.6 |
| Jamesbond | 27 | 302.8 | 320.0 | 625.0 | 1995.0 | 694.0 | 480.0 | 657.4 | 450.0 |
| Kangaroo | 54 | 3,035.0 | 660.0 | 9870.0 | 13440.0 | 14398.2 | 14720.0 | 8548.1 | 10680.0 |
| Krull | 1,566.59 | 2,665.5 | 9191.1 | 9366.9 | 9918.7 | 9293.1 | 8732.7 | 6069.8 | 9549.0 |
| KungFuMaster | 451 | 22,736.3 | 62800.0 | 55060.0 | 36020.0 | 27786.9 | 36940.0 | 28394.6 | 42600.0 |
| MontezumaRevenge | 0.0 | 4,753.3 | 1.0 | 1.0 | 1.0 | 0.0 | 1.0 | 0.0 | 0.0 |
| MsPacman | 242.6 | 6,951.6 | 3230.0 | 2168.0 | 2673.0 | 2557.7 | 2568.0 | 3244.8 | 2568.0 |
| NameThisGame | 2404.9 | 8,049.0 | 4702.0 | 6278.0 | 11739.0 | 11161.7 | 12394.0 | 10859.2 | 9200.0 |
| Phoenix | 757.2 | 7,242.6 | 5398.0 | 12043.0 | 12324.0 | 21813.7 | 32086.0 | 30561.7 | 18558.0 |
| Pitfall | -265 | 6,463.7 | 1.0 | 1.0 | 1.0 | -37.4 | 1.0 | -82.8 | 0.0 |
| Pong | -20.34 | 14.6 | 20.0 | 20.7 | 20.8 | 20.5 | 20.9 | 20.3 | 21.0 |
| PrivateEye | 34.49 | 69,571.3 | 100.0 | 100.0 | 100.0 | 62.1 | 100.0 | 284.2 | 100.0 |
| Qbert | 188.75 | 13,455.0 | 8150.0 | 16575.0 | 13830.0 | 12307 | 15782.5 | 13325.7 | 6530.0 |
| RiverRaid | 1575.4 | 17,118.0 | 8350.0 | 10232.0 | 8714.0 | 10102.4 | 9350.0 | 10975.4 | 11998.0 |
| RoadRunner | 7 | 7,845.0 | 44950.0 | 54490.0 | 54620.0 | 43574.9 | 42530.0 | 37909.9 | 52600.0 |
| Robotank | 2.24 | 11.9 | 13.2 | 22.5 | 48.1 | 52.4 | 34.4 | 48.4 | 48.1 |
| Seaquest | 88.2 | 42,054.7 | 1444.0 | 10666.0 | 2640.0 | 4424.3 | 11685.0 | 1745.2 | 14795.0 |
| Skiing | -16267.9 | -4,336.9 | -13340.4 | -19040.3 | -29970.3 | -27543.7 | -8983.3 | -23701.6 | -29970.3 |
| Solaris | 2346.6 | 12,326.7 | 582.0 | 192.0 | 956.0 | 1220.5 | 3336.0 | 996.8 | 792.0 |
| SpaceInvaders | 136.15 | 1,668.7 | 1005.0 | 1725.5 | 1826.5 | 1794.6 | 1216.0 | 991.8 | 2302.5 |
| StarGunner | 631 | 10,250.0 | 1270.0 | 22600.0 | 38380.0 | 52937.6 | 52050.0 | 46658.3 | 43820.0 |
| Tennis | -23.92 | -8.3 | -5.7 | -1.5 | -11.9 | -11.3 | -1.5 | -13.7 | 13.3 |
| TimePilot | 3682 | 5,229.2 | 1420.0 | 3260.0 | 6030.0 | 6338.7 | 7900.0 | 4516.2 | 7060.0 |
| Tutankham | 15.56 | 167.6 | 206.6 | 186.0 | 178.3 | 153.8 | 205.2 | 190.5 | 202.8 |
| UpNDown | 604.7 | 11,693.2 | 19145.0 | 16046.0 | 17074.0 | 14402.2 | 44746.0 | 19350.3 | 20063.0 |
| Venture | 0.0 | 1,187.5 | 1.0 | 1.0 | 1.0 | 3.9 | 1.0 | 39.3 | 1370.0 |
| VideoPinball | 15720.98 | 17,667.9 | 270050.9 | 477206.8 | 388106.7 | 44599.9 | 288137.2 | 152686.0 | 164597.3 |
| WizardOfWor | 534 | 4,756.5 | 1440.0 | 1620.0 | 4890.0 | 8670.1 | 4480.0 | 5095.3 | 3250.0 |
| YarsRevenge | 3271.42 | 54,576.9 | 12507.9 | 15954.4 | 17593.8 | 12336.1 | 8516.8 | 13420.7 | 13507.3 |
| Zaxxon | 8 | 9,173.3 | 1.0 | 5910.0 | 7410.0 | 9077.0 | 4640.0 | 8557.9 | 10320.0 |

Table 3: Scores of all algorithms averaged over 3 seeds across 55 Atari games.

