# OpenReview forum: "Distributional Reinforcement Learning via Sinkhorn Iterations"
_ICLR.cc/2023/Conference — Submitted to ICLR 2023_

### Official Review · Reviewer_tLLY · 2022-10-24

**Confidence:** 3
**Correctness:** 3
**Technical Novelty And Significance:** 2
**Empirical Novelty And Significance:** 2
**Recommendation:** 3

**Clarity, Quality, Novelty And Reproducibility:**

As mentioned above, the clarity could be improved.
- There are typos throughout such as “Sinkhron,” “suit,” etc.
- The distr bellman operator should really not be an =. It should be distributionally equal (this should be apparent from the notation, not just the text).
- theta and theta_* are not defined in Sec 3.1 or Algorithm 1.
- It’s only described very informally how the Z_{\theta} are represented.
-  (5) and (6) should not have k in Q and Z.
- Generally there are just arbitrary mixtures of treating the $Z$ (and related quantities) as either random variables or as distributions.
- The actual algorithm is never explicitly stated. Just the update rule is given. It’s unclear how the data is collected and used in the update rule. It’s never defined what the policy should do.
- Algorithm 1 returns a distance. It’s unclear what one is supposed to do with this.
- The constraints in (13) seem to be inaccurate. I think they should sum to the input distributions.
- Proposition 1 “can be equivalent to.” Is it only sometimes equivalent or always? If sometimes, under what conditions? I think this should be made more precise.


**Strength And Weaknesses:**

Strengths:
- The update rule appears to be novel for distributional RL; however, it is already known that you can use any distance. This is just one particular example that has also seen applications and success in other areas.
- The experiments are fairly extensive, evaluating the algorithm against existing DRL methods across 50+ Atari tasks
- The theory seems reasonable, but I didn’t check carefully.

Weaknesses:
- Although probably new in application, the algorithmic idea is not particularly novel, so the results should probably be judged more on significance of improvement and analytical insights.

- The clarity could be greatly improved. There are numerous inconsistencies, typos, undefined notation, etc. See specifics below.

- The novel part of Theorem 1 (c) is not actually a contraction in the usual sense as it depends on the iterates. It’s not clear that this is truly significant and actually yields the desired convergence.

- The rates in Table 1 are not proven (there is no theorem) or discussed or interpreted. This is especially important since the first one is convoluted.

- The empirical results do not appear to yield that significant of an improvement over existing algorithms. But there is improvement in some domains.

- There are claims made in the experiments section that don’t appear to be supported by the results. It says “SinkhornDRL achieves better performance across more than half” but it looks like it is the opposite. It’s really only half the games for (a) and less than half for (b). Am I just reading the plot backwards? Blue (negative) is worse for Sinkhorn, no?

“the performance of SinkhornDRL tends to QR-DQN or MMDDRL on Seaquest when we decrease or increase epsilon.” This doesn’t seem to be supported by the plots. The performance decreases, but it’s not at all clear that it’s leveling out at the performance of the other algorithms.



Questions:
- Why was DQN not included in the ratio plots?

- In Fig 3 (b), since the performance doesn’t seem to be effected much by N, why not choose N to be very small such as N = 1, 2? Theoretically it doesn’t make much sense, but it seems it would be useful to see when performance finally drops off.
- “Sample complexity” in Table 1 is never defined. Sample complexity to achieve what? What is the objective? Sub-optimality in distribution, in expectation, etc?


**Summary Of The Paper:**

This paper introduces a new method for solving distributional RL problems. The main idea behind the paper is to use the Sinkhorn distance (wasserstein with entropic regularization) as the basis for the distributional updates. Several theoretical results are proven concerning the contractive property of the Sinkhorn update as well as equivalences with other updates in certain settings. Extensive experimental results are presented with comparison to existing distributional RL algorithms

**Summary Of The Review:**

In summary, this paper presents an interesting idea and does an interesting theoretical analysis and extensive experimental evaluation. However, it seems to fall short in a few categories as described in the above sections, with several unanswered questions. The lack of clarity is also a drawback.

Other notes:
- More exact convergence results exist for the general sinkhorn algorithm: “Near-linear time approximation algorithms for optimal transport via Sinkhorn iteration”
- I encourage the authors to also consider more in depth analysis of experimental results as prescribed by the following paper considering only 3 seeds are used: "Deep Reinforcement Learning at the Edge of the Statistical Precipice"

---

> ### Author Response · Authors · 2022-11-08
> **Author Response**
>
> Thank you for your valuable comments and suggestions. Below are our responses to your concerns. Please be free to let us know if there are any further questions.
>
> ### Weakness
> * From distributional RL literature, based on the fact that Wasserstein distance and MMDDRL are two dominant classes, Sinkhorn, which interpolates them, is definitely worthwhile to study. A non-trivial contraction proof in Theorem 1(3) and regularized MMD correlation are our theoretical contributions. Experiments on 55 Atari games with learning curves of all considered algorithms provided are extensive.
> * Thanks for the clarify suggestions. We improved them in the revised version. Please check it.
> * We would like to clarify your concern and emphasize that Theorem 1(c) is indeed a contraction **in the usual sense regardless of iterates**, which is guaranteed in 3.3 the new contraction mapping theorem and 3.4. The key reason is that $ \Delta^{U, V}(a, \alpha) $ in Eq.27 is strictly less than 1, although it is non-constant and depends on iterates. This proof is rigorous, novel and non-trivial. The superior performance of Sinkhorn corroborates its favorable contraction property.
> * As we have stated in Sec 3.2, “Theoretical results regarding Sinkhorn divergence is based on *Genevay et al.2019* and the detailed convergence proof of other distances is also provided in Appendix B.” Since results in Table 1 are mainly summarized from the existing paper, we mainly provide them for reference and a descriptive comparison.
> * To be best of our knowledge, no distributional RL algorithm can beat others across all Atari games. SinkhornDRL is significantly better than MMDDRL on a proportion of games, and is competitive on most games. **In Table 2 of Appendix E, SinkhornDRL outperforms MMDDRL according to the mean Human normalized score**.
> * Thanks for pointing out this clarity issue. We apologize it is slightly inaccurate and it should be a half or nearly a half. However, we need to emphasize both Sinkhorn and MMDDRL are close (blue and orange bars with small heights) in most games, SinkhornDRL is significantly better in a small portion of games. We have revised them in the current version.
>
> ###  Question
> * Sinkhorn performs much better than DQN, and we focus on the comparison with the most theoretically relevant QRDQN and MMDDRL. We can immediately show the ratio improvement over DQN if needed.
> * As you require, we provide this sensitivity analysis w.r.t. N in Figure 11 of Appendix G.1. It suggests that a small N leads to divergence in Seaquest while may maintain a desirable performance on the easier tasks, e.g, Breakout.
> * We mentioned the definition of sample complexity in Sec 2.2 MMD part, i.e., i.e., approximating the distance with samples of measures. This is also discussed in MMDDRL Theorem 3 and Proposition 2.
>
>
> ### Clarity, Quality, Novelty And Reproducibility
> * Thanks. We fixed typos.
> * Thanks for this suggestion. We added that to the equation.
> * We made more clear definitions in the revised paper.
> * We add a formal definition in the related works section of Appendix A.
> * We argue that K should be added in the k-th Neural FQI or FZI.
> * We sometimes mix the use of Z just for easier understanding. This random-variable definition of the distributional Bellman operator is appealing and easily understood due to its concise form, although its value-distribution definition is more mathematically rigorous [1].
> * We follow the MMDDRL update rule and omit the general algorithm part for brevity.
> * We follow the MMDDRL update rule algorithm 1. The evaluation of the returned distance is by Sinkhorn iterations in Algorithm 2.
> * We follow SinkhornGAN paper, and it should be the exact form we present currently.
> * ’is equivalent to’ is more accurate. We revised it in the current version.
>
> ### Other notes.
> * We thank you for your recommendation. We read it and cited this paper at the convergence of Sinkhorn iteration part.
> * We agree that more seeds are more convincing as pointed out in the referred paper and we will do more rigorous experiments in the future. For the current paper, we strictly follow the previous QRDQN,C51 and MMD averaged on 3 seeds, and thus we argue that it is still strong enough for publication.
>
> [1] Marc G. Bellemare, Will Dabney, and Mark Rowland. Distributional Reinforcement Learning. MIT Press, 2022. http://www.distributional-rl.org.

---

> > ### Comment · Reviewer_tLLY · 2022-11-24
> > **Thanks**
> >
> > Thanks for the response. Re the contraction: I still do not think this statement or proof are correct yet.
> > 1. First there is a typo in Theorem 2. The distance $d$ is not a contraction, it is a distance. This should say $g$.
> > 2. $g$ is also not a contraction by this definition. A contraction $g$ defined on $(M, d)$ satisfies $d(g(x), g(y)) \leq c d (x, y)$ for a constant $c \in [0, 1)$ for all possible $x, y \in M$ (https://en.wikipedia.org/wiki/Contraction_mapping).
> > 3. $g$ could be a *contractive* operator, but this does not retain the desired properties of a contraction mapping theorem, which Theorem 2 tries to resolve.
> > 3. The proof of Theorem 2 asserts $\prod_{i = 1}^k q_{i, i -1}  \rightarrow 0$. I do not think this is true without further assumptions. Consider $q_{i, i - 1} = (1 - 1/(i + 2)^2)$. This satisfies $q_{i, i - 1} < 1$ but $\prod_{i = 1}^k q_{i, i -1} > 1/4$ for all $k$.

---

> > > ### Author Response · Authors · 2022-11-24
> > > **Further Author Response**
> > >
> > > Thanks for raising new questions.
> > >
> > > **1.** We apologize for this typo and we will fix it in the revised version.
> > >
> > > **2&3.**  As you said, g may not satisfy a **classical** contraction mapping definition in terms of a constant, but g can be a contractive operator **while retaining the desired properties**. In particular, we show that $X^{(k)}$ can tend to a unique $X^*$ in distribution with respect to d as $k\rightarrow +\infty$, which are the main properties stated in the classic contraction mapping theorem (Theorem 1 in Section 1.2) [link](https://terpconnect.umd.edu/~petersd/666/fixedpoint.pdf), to the best of our knowledge.
> > >
> > > **4.** We thank you for raising this rigorous mathematical question. If we are wrong, please be free to let us know, but here we are trying to explain it within our knowledge in the following way. Assume that $i$ could equal $+\infty$. Based on Eq.27, we know $q_{i, i-1}<1$ for $i=1,...,+\infty$. Denote $q_{\text{max}}=max\\{q_{i, i-1}\\}_{i=1}^{+\infty}<1$. According to the fact that real numbers are dense, there always exists a uniform  $\epsilon>0$ such that
> > >
> > > $q_{\text{max}} \leq 1-\epsilon<1$. Thus $\Pi_{i=1}^{k}q_{i, i-1} \leq (1-\epsilon)^k \rightarrow 0$ as $k\rightarrow +\infty$. In your example, $q_{i,i-1}=1$ when $i=+\infty$, which does not satisfy our condition. People may argue that $i$ can not equal $+\infty$. In this case, we agree with you we need to add an assumption: there exists a universal $\epsilon>0$ such that $q_{\text{max}} \leq 1-\epsilon<1$, which we do believe is mild. By contrast, in practice, your particular monotonically increasing sequence is very unlikely to happen, but we appreciate you raising this rigorous mathematical problem.
> > >
> > > We hope our response could be helpful for you to better understand our work. Thanks a lot.

---

> > > > ### Comment · Reviewer_tLLY · 2022-12-09
> > > > **Minor point**
> > > >
> > > > Thanks again for your response.
> > > >
> > > > For general sequences $q_{i ,i - 1}$, $q_{\max}$ is not defined. Consider $q_{i, i - 1} = 1 - 1/i$ or the example that I gave in the previous response. There is no element that is at least the value of all other elements, so the part about there existing such an $\epsilon$ is not necessarily true. The $\sup$ is defined, but it is just equal to 1.
> > > >
> > > > I agree that if there was an assumption stating you only consider sequences for which such an $\epsilon$ exists, then the proof would be fine. However if that assumption held, then this would be a typical contraction mapping.

---

> ### Comment · Area_Chair_VpBE · 2022-11-20
> **Any comments to the responses from authors?**
>
> Dear Reviewer tLLY,
>
> Thank you very much for your informative review.  The authors have provided clarifications and responses to your concerns.  How did they change your evaluation?

---

### Official Review · Reviewer_LQMW · 2022-10-25

**Confidence:** 4
**Correctness:** 4
**Technical Novelty And Significance:** 2
**Empirical Novelty And Significance:** 2
**Recommendation:** 5

**Clarity, Quality, Novelty And Reproducibility:**

The writing is clear and the proposed algorithm is novel and high quality. I am unsure about the reproducibility, in particular because the OT requires a high implementation complexity, but from a theoretical perspective, it should be reproducible given how the proposed algorithm only replaces the divergence used in distributional RL.

**Strength And Weaknesses:**

Strengths:
- The paper provides a nice theoretical framework for distributional RL via sinkhorn iterations and how the optimal transport problem it solves in distributional RL interpolates between Wasserstein and MMD distances, both of which are appealing for their contractive divergences in Bellman equations. Furthermore, the free choice of cost function is an interesting one which allows more flexibility than the contractive requirement for divergences needed for conventional distributional RL. It is shown that sinkhorn algorithm using Gaussian kernels is equivalent to regularized MMD.
- There is an implementation to utilize the sinkhorn algorithm for distributional value-distribution learning in the RL setting, which requires Sinkhorn to be differentiable and efficiently computed.
- For several tasks, the proposed algorithm significantly outperforms a state-of-the-art distributional RL algorithm in MMDRL, which is impressive.

Weaknesses:
- The advantages of OT vs MMD is known to be mostly taking advantage of the data geometry and the geometry of the flat distance measures when following the gradient of the divergences. However, it is unclear to me whether this advantage is applicable to the setting of estimating a 1-dimensional value-distribution. Furthermore, OT suffers from high computational cost, with the same number of design choices required to tune (cost function replacing kernel choice) as MMD.
- Along the lines of the above, it is unclear whether the benefits of this algorithm outweigh its implementation difficulty along with overall complexity. The results for which MMDRL is superior seems to indicate that for a majority of Atari games, this does not do much better
- It would be have interesting to see if these results generalize to harder tasks either in discrete control or even continuous control.

**Summary Of The Paper:**

This paper proposes the sinkhorn algorithm for optimal transport to learn the distributional Q-estimate via a bootstrapped wasserstein distance between distributions. It builds on top of existing distributional off-policy model-free RL methods which estimate a distribution over the discounted sum of rewards ahead and leverage it for policy learning. Additionally, the paper proposes a theoretical framework wherein their method interpolates between the wasserstein distance and MMD between distributions.

**Summary Of The Review:**

Overall, while this is a nice applied paper, the Sinkhorn algorithm used in this context requires further study, in particular why it would work well for certain tasks and not others. I would recommend rejecting the paper in its current state but would increase my score if more justification could be provided for how beneficial this approach is vs existing ones for other tasks.

---

> ### Author Response · Authors · 2022-11-08
> **Author Response**
>
> Thank you for your valuable comments and suggestions. Below are our responses to your concerns. Please be free to let us know if there are any further questions.
>
> ### Weakness 1. Advantages of OT vs MMD
> Although the resulting value distribution is 1-dimensional, the geometry behind the high-dimensional states might be complex **as states might be with high probability concentrated in the vicinity of some underlying manifold with a low dimensional (>1) space**. Thus, OT, which leverages data geometry, is still worthwhile to study in the context of distributional RL, in which MMD is inferior in this aspect. This can also be empirically verified as MMDDRL is worse than OT-based algorithms in certain environments, e.g., WizardOfWor in Figure 1.
>
> In OT-based literature, **Sinkhorn divergence is widely accepted as it can efficiently evaluate OT by introducing entropic regularization, largely mitigating the costly computation drawback of OT**. This slight increase in overhead can be tolerant when we have access to more computational resources. We mentioned in Section 5.2 that SinkhornDRL slightly increases the overhead as verified in Appendix G. Meanwhile, the resulting SinkhornDRL introduces unrestricted statistics, i.e., samples, to represent the value distribution, which is more expressive than quantile-based distributional RL algorithms that additionally suffer from non-crossing issues.
>
> ### Weakness 2. Implementation Difficulty.
> We provide the reproductive code in the supplementary file and you can check *deep_rl/agent/SinkhornDQN_agent.py* for our efficient implementation, which is indeed not very complex. The sinkhorn iteration is implemented based on the reliable open-source code of Sinkhorn-GAN in https://github.com/gpeyre/SinkhornAutoDiff.
>
> As shown in Algorithm 1, the major difference between MMDDRL and SinkhornDRL is we replace the MMD with Sinkhorn iterations that have been widely accepted in practice. **This implies that SinkhornDRL is easy to implement and also only slightly increases the cost** as suggested in Figure 13 of Appendix F.2.
>
> As shown in Figure 2(b), SinkhornDRL is competitive with MMDDRL in most Atari games, but SinkhornDRL **significantly** outperforms MMDRL in a large proportion of games, including Atlantis, Zaxxon and Tennis. All learning curves are also provided in Appendix.
>
> ### Weakness 3. Harder Task.
> As we strictly follow the evaluation standard of existing distributional RL papers, e.g., QRDQN, C51 and MMD, our experiments are conducted on 55 Atari games, which is already heavy work. Thus, we argue our empirical evaluation is sufficient for a publication, although results on more tasks, as you suggested, can be more convincing.

---

> ### Comment · Area_Chair_VpBE · 2022-11-20
> **Any comments to the responses from the authors?**
>
> Reviewer LQMW,
>
> Thank you very much for your informative review.  The authors have provided responses to your concerns.  How did they change your evaluation?  Did they serve as justification for the benefit over existing methods?

---

### Official Review · Reviewer_VKzt · 2022-11-03

**Confidence:** 4
**Correctness:** 2
**Technical Novelty And Significance:** 2
**Empirical Novelty And Significance:** 1
**Recommendation:** 3

**Clarity, Quality, Novelty And Reproducibility:**

**Clarity**:  moderate

The paper is well organized, clearly written and easy to follow.

**Quality**:  moderate

Though the writing is okay, the quality is limited because SinkhornDRL comes with limited technical novelty compared with MMDDRL,  there is not inclusive discussion/comparison with related distributional RL methods,  and the empirical evaluation setting/result is not convincing, while sine if the result even highlights SinkhornDRL is less effective than baselines.

**Novelty**:  weak

The novelty is weak because as the authors claim, SinghonDQN is only different from MMDDRL with their distribution divergences.  And the only new theoretical novelty is the proof of contraction theorem.

**Reproducibility**: weak

The authors present very limited details on the implementation details of the method. Though the authors claim the empirical setting is inherited from QRDQN, it is possible settings as both sides are not identical so the reproducibility information for this work needs to be self-contained.

**Strength And Weaknesses:**

**Strength**
* Sinkhorn divergence has not been formally considered for distributional learning in the existing distributional RL literature.
* The paper is clearly written and easy to understand.

**Weaknesses**

[Motivation]
* Overall I feel this work is not well motivated from distributional RL perspective. The motivations  introduced for the work are mainly related to: (1) introducing a new distributional RL algorithm  (2) SinkhornDRL bridges between Wasserstein distance and MMD. It remains unclear what challenge in distributional RL could be  tackled by Sinkhorn, and why bridging the two major types of distributional attempts could be more beneficial than employing each single one. (I think motivating the RL perspective from the entropy-regularization might also be valid but there is little statement of this type.)
* The related works discussed/compared in this paper are not inclusive enough. In the sections before methodology, I only see two major distributional RL methods, QR-DRL and MDDRL,  introduced in details, while it would be unfair to describe the literature through only two works. In some important part in methodology part. the authors also compare SinkhornDRL with a limited subset of distributional RL baselines, e.g., the complexity and convergence rate shown in Table 1. I refer the authors to the related works presented in the MDDRL paper, where there is a more inclusive and informative categorization/description on related works.


[Formulation]

* The overall SinkhornDRL solution comes with limited novelty because the algorithmic approach of solving moment matching with unrestricted statistics of deterministic samples highly depends on the existing work MMDDRL. The main change of SinkhornDRL from  MMDDRL is very small, i.e., to minimize a regrarized squared MMD, while the major idea like applying unrestricted statistics and approximating the distance with samples of measures, are the same.
*  The statistical relationship between Sinkhorn and MMD is not newly developed, since it's well adopted from existing statistical literature.
* The authors make rather strong claim on the theoretical contribution of this paper, but I feel it is not very convincing as the only new proof is the proof of contraction theorem only, which itself is not very complicated. Furthermore, considering the limited algorithmic change has not been well justified by the empirical evaluation, I feel the significance of the proposed method is not enough for being published in ICLR conference.

[Empirical Evaluation]

* The authors only adopt a single experimental domain Atari 2600, but they could not align the well adopted Atari 2600 setting with the other baselines which all adopt the same domain, e.g., C51, QRDQN and even MMD. The authors only adopt a much smaller frame number, which they claim is **10M**, 1/20 from the most widely adopted standard **200M**.  I understand the computational resource required for Atari 2600 is a lot, but if not accessible to such resources, should turn to alternative domains which have lots of candidates (e.g., Mujoco and ViZDoon), as well as proof-of-the-concept toy domains. To me, lack of the computational budget is not a valid excuse for using such small frame budget and make the setting biased from its most related baselines.

* There is one important issue the experiment, which is the **10M** frame budget the authors claim they use. $\textcolor{red}{\mbox{I suspect the authors actually use a much larger number than 10M}}$. The authors never mention *frame repeat* or  *frame stack* in their paper, so based on the well-adopted standard, **10M** should correspond to the total number of frames, not that number before multiplying frame repeat. So it's 1/20 to C51, QRDQN, MMDDQN and many other classic DRL methods like IMPALA, RAINBOW and MUZERO. It's important to make that much progress within 10M only. For example, with Breakout, SinkhornDRL uses less than 5M frames to score > 350, but that progress in C51 paper's Fig 3, QRDQN paper's Fig 6 and even Rainbow's Fig 5, it takes way more than 10M frames for Breakout agent to progress to that standard. [Highly recommend reviewers and AC to take a look into aforementioned papers].

* I think the reported median/mean HNS scores for Sinkhorn as well as its baselines (in Table 2) are not responsible. I have computed the median HNS for C51 and QR-DQN from a very reliable  open-source experiment logs released in a public repository dqn-zoo. Within 10M frame, the median HNS for both methods are bellow 50%, and within 10M * 4 repeat (assume they forget to multiply frame repeat),  the scores for both methods > 100%. For neither case, the standards of median HNS aligns with the reported numbers. Therefore, I'm not convinced if SinkhornDRL could be identified as an effective distributional RL method. $\textcolor{red}{\mbox{Even their reported numbers for Sinkhorn underperform MMDQN in both median and mean HNS metrics. }}$
It seems that the empirical results for their sole benchmark domain reveals the performance of MMDQN is better than Sinkhorn.

* In Fig 2, the ratio improvement (%) curve seems to show that both QRDQN and MMDDRL could dominate Sinkhorn in the number of games they outperform Sinkhorn, i.e., width for orange bars are shorter than blue bars.

* The baselines are not inclusive. Note that both C51 and QRDQN are not methods with the unrestricted distributions like SinkhornDRL. Instead, their follow-up yet well-adopted works IQN and FQF are methods with unrestricted distributions. I feel the authors should include those unrestricted distribution baselines (apart from MMDQN) for compression.

**Summary Of The Paper:**

This paper proposes a new distributional RL algorithm named SinkhornDRL, which proposes to solve the distribution matching problem in distributional RL with Sinkhorn divergence.

Sinkhorn divergence in formulation is equivalent to an entropy-regularized version of MMD, and it could be featured as an interpolation between Wassersteon distance  and MMD, so that it simultaneously leverages the geometry of Wasserstein distance and unbiased gradient estimate property of MMD in moment matching.  SinkhornDRL is highly related to MMDDRL, as they are both moment matching approaches that use samples to approximate the distribution associated with kernel tricks.

The main new theoretical insight introduced in this paper is to prove the contraction property of SinkhornDRL in a short section from Appendix (Equation 32 in Appendix 3.4).  The other proof on the statistical properties of Sinkhorn, such as its equivalence form to regularized MMD, is not novel.

For empirical comparison, the authors compare SinkhornDRL with DQN, C51, QRDRN and MMDDQN on their modified 55 Atari games benchmark, where they evaluate the agents under 10M frame regime. The results show that SinkhornDQN could outperform MMDDQN occasionally and overall results in inferior median and median HNS than MMDDQN.



**Summary Of The Review:**

This paper introduces a novel distributional RL method where the distributional divergence is estimated from deterministic samples with Sinkhorn divergence. The authors evaluate the method on a Atari 2600 domain under a customized frame number 10M, where I feel the results reveal MMDDRL outperforms SInkhornDRL and the settings are falsely claimed.

I'll re-evaluate the recommendation score to this paper if  my concerns on (1) technical novelty, (2) presentation on empirical settings and results,  (3)  sufficient details, atari wrappers, code or pseudocode  to implement the experiment could be provided to convince me the result is reproducible. It's even better if the authors could replace the results under 10M with those under 200M.

---

> ### Author Response · Authors · 2022-11-08
> **Author Response 1**
>
> Thank you for your valuable comments and suggestions. We give a clearer explanation about the motivation, but we respectfully disagree with some formulation points you mentioned. We apologize for the confusion and mistake in the evaluation and we revised the empirical results according to facts. All issues are fixed in the current version. Please be free to let us know if there are any further questions.
>
> ### [Motivation]
> * In fact, in this paper we have shown the key challenge in distributional RL is **the choice of $d_p$ as well as the corresponding representation way of $Z_\theta$ in Section 3.2**. Thus, with both appealing properties, Sinkhorn divergence is well motivated as a new distance that exactly interpolates Wasserstein distance and MMD, which correspond to the current two dominant distributional RL families. **SinkhornDRL is supposed to play an integral part in the whole distributional RL literature**.
>
> * Thanks for this suggestion. In the original version, we in fact put related works in Section 3.2 in the framework of Neural FZI, but it seems not inclusive enough as you pointed out. Hence, we additionally add a related work section in Appendix A based on but more detailed than the related works section in MMDDRL. Please refer to Appendix A.
>
> ### [Formulation]
> * As a regularized Wasserstein distance, Sinkhorn is in fact motivated to efficiently solve Wasserstein distance and thus can be applied in distributional RL algorithms. Due to the fact that MMDDRL is the SOTA algorithm and Sinkhorn interpolates with Wasserstein distance and MMD in nature, we further establish Sinkhorn’s connection as a regularized MMD as a theoretical contribution. Unrestricted statistics should obey the choice of distribution divergence, and thus both Sinkhorn and MMD leverage the advanced deterministic samples. We need to clarify that **Sinkhorn is fundamentally different from MMD as it is in fact a kind of optimal transport distance or entropic regularized Wasserstein distance to be more specific**.
>
> * We respectfully disagree on this point as our regularized MMD result in Proposition 1 is based on our own derivation and MMDDRL paper without referring to any other statistical literature. **To the best of our knowledge, this connection is novel, non-trivial, and NOT established yet in the existing statistical literature**. If there is, please point it out and we will be grateful.
>
> * We would like to clarify that **3.1, 3.2, 3.3 and 3.4 are all new proof in a whole proof framework**. For example, it takes us a long time to derive the scale-sensitive property in 3.2, which is definitely non-trivial. Motivated by the non-constant factor $\Delta^{U, V}$, we thus propose a new contraction mapping theorem based on the series convergence lemma. Finally, we arrive at the eventual contraction proof in 3.4.

---

> ### Author Response · Authors · 2022-11-08
> **Author Response 2 (continued)**
>
> ### [Empirical Evaluation]
> * **10M Issue**. We apologize for this misunderstanding. **10M should refer to time steps, which correspond to 40M frames (1/5 to 200M frames)**. Thus, this fact should align with previous results in C51, QRDQN, and MMDQN mentioned by you. The reason why we focus on 40M frames is that we are trying to provide **trustworthy results** on standard Atari games, especially **learning curves** of all typical 5 distributional RL algorithms as suggested in Figures 4 to 9, which have not been exhibited in the previous papers. Note that learning curves across 55 Atari games are more informative and reliable than just HNS with Mean and Median as we find mean is sensitive to games with high-level returns, and the median is overly robust. We hope you can understand all of our current results have already been a huge computation cost for us.
>
> * **HNS Issue**. We really thank you for pointing out this problem. Firstly, we apologize as we find the previous HNS was erroneously computed by *algorithmscore/humanscore*, while forgetting to consider the random play score in the computation. We addressed this issue and update the correct ones in Table 2 of Appendix E in the current version. **We also additionally provide averaged scores of all algorithms in Table 3 of Appendix H towards trustworthy reproductivity**. Notably, the median of HNS for almost all distributional RL algorithms is greater than 100%, which aligns with your experience. **It also turns out that in the revised result, SinkhornDRL outperforms MMDDRL in mean HNS, verifying our superiority**. Please refer to Table 2 of Appendix E for more details.
>
> * **Experimental setting**. Previous results are implemented by TensorFlow Dopamine framework, while our Pytorch implementation is based on a reliable open-source GitHub [1] with 2.8k stars. In table 2, we provide both results from us and TensorFlow calculated from the released code of MMDDRL[2] after the same time steps. It suggests the superiority of SinkhornDRL over MMDDRL in the mean HNS.
>
> * **Raito improvement in Figure 2**. Firstly, SinkhornDRL significantly outperforms both QRDQN and MMDDRL in a large number of games (albeit less than half). Also, SinkhornDRL is competitive with MMDDRL on the most of remaining games as most blue bars are very short and close to 0.
>
> * **IQN and FQF**. We would like to clarify that IQN and FQF extend QRDQN by increasing the model expressiveness, but they are still using quantiles rather than unrestricted statistics. **From the perspective of distribution divergence, QRDQN and MMDDRL are direct counterparts for SinkhornDRL**. As discussed in MMDDRL, IQN and FQF can extend algorithms with unrestricted statistics naturally. For example, we can use the proposal network in FQF to learn the weights of each Dirac component in MMDQN instead of using equal weights 1/N.
>
> ### [Technical Novelty, Empirical Results and Reproductivity]
> Note that all distributional RL algorithms mainly differ from the choice of distribution divergence along with the corresponding distribution representation manner of $Z_\theta$. In the general framework of Algorithm 1, SinkhornDRL seems similar to MMDDRL as both are sample-based, but **Sinkhorn divergence is fundamentally different from MMD as it is in fact seen as a solvable Wasserstein distance via Sinhorn iterations**. Also, our theoretical proof is novel rather than adapted from existing literature.
>
> Empirically, SinkhornDRL outperforms MMDDRL on a large number of games, although less than half, while is competitive on most games as shown in Figure 2.
>
> We provide the source code in the supplementary materials adapted from [1]. Results are trustworthy as we also provide all learning curves as well as final scores in Appendix H. We thank you for all your suggestions and hope all our responses can clarify your concerns.
>
> **Reference**
>
> [1] Shangtong Zhang. Modularized implementation of deep rl algorithms in pytorch. https://github.com/ShangtongZhang/DeepRL, 2018.
>
> [2] Nguyen-Tang. Distributional reinforcement learning via moment matching. https://github.com/thanhnguyentang/mmdrl, 2021.

---

> ### Comment · Area_Chair_VpBE · 2022-11-20
> **Any comments to the feedback from authors?**
>
> Dear Reviewer VKzt,
>
> Thank you very much for your detailed review.  The authors have corrected a few mistakes and provided responses to your concerns.  How did they change your evaluation?

---

### Official Review · Reviewer_vqCE · 2022-11-04

**Confidence:** 5
**Correctness:** 3
**Technical Novelty And Significance:** 3
**Empirical Novelty And Significance:** 2
**Recommendation:** 5

**Clarity, Quality, Novelty And Reproducibility:**

The paper is clearly written.

I consider it as a "new" work. The new contributions of this paper is to replace the divergence in DRL to be Sinkhorn divergence, which can be viewed as an interpolation between Wasserstein distance and MMD.

I am not convinced with average over only 3 seeds. At least a few experiments should be presented with more number of seeds.

**Strength And Weaknesses:**

Strengths: 1. The new contributions of this paper is to replace the divergence in DRL to be Sinkhorn divergence, which can be viewed
as an interpolation between Wasserstein distance and MMD.

Weakness: 1. "The main limitation of our proposal is that the superiority over existing state-of-the-art algorithms
may not be sufficiently significant." The authors themselves point this out. So what is the advantage of using Sinkhorn iterations?
2. Figure 3 (b) Why does the performance degrade with increase in sample size? Increasing the samples should better approximate the return distribution.


**Summary Of The Paper:**

This paper proposes a new DRL method leveraging Sinkhorn divergence. Analogous to DQN, they consider learning the action value distribution by minimizing the divergence between the current distribution and the target distribution. The main difference with previous DRL methods is they replace the divergence to be Sinkhorn divergence. Theoretically, they study the convergence of the distributional
Bellman operator, the moment matching meaning of Sinkhorn divergence, and the connection with MMD DRL. Although the paper is generally well-written, the experimental results does not show significant improvements over the current state-of-the art methods.

**Summary Of The Review:**

This paper proposes a new DRL method leveraging Sinkhorn divergence. Analogous to DQN, they consider learning the action value distribution by minimizing the divergence between the current distribution and the target distribution. The main difference with previous DRL methods is they replace the divergence to be Sinkhorn divergence. Theoretically, they study the convergence of the distributional
Bellman operator, the moment matching meaning of Sinkhorn divergence, and the connection with MMD DRL. Although the paper is generally well-written, the experimental results does not show significant improvements to the current state-of-the art methods.

---

> ### Author Response · Authors · 2022-11-08
> **Author Response**
>
> Thank you for your valuable comments and suggestions. Below are our responses to your concerns. Please be free to let us know if there are any further questions.
>
> ### Weakness 1. Significant Improvement in experiments.
>
> Based on Figure 2 (b), Sinkhorn outperforms MMDDRL **significantly** on a large number of games albeit not a half, especially on Tennis, Atlantis and Seaquest as shown in Figure 1. Also, Sinkhorn is **competitive** on the majority of the remaining games as most games in blue are close to 0. It should also be noted that there is no algorithm that can beat other baselines across all Atari games due to the huge agnostic environment complexity on Atari games. Similar evidence can refer to Figure 4 in MMDDRL paper.
>
> We need to emphasize that the proposal of SinkhornDRL has its technical contribution, and appealing theoretical results with a non-trivial contraction proof in Theorem 1 and connection with regularized MMD in Proposition 1, as well as its superior or competitive performance with the SOTA algorithm. Due to the fact that SinkhornDRL interpolates between two dominant distributional RL classes, based on Wasserstein distance and MMD respectively, **SinkhornDRL is supposed to play an integral part in the whole distributional RL literature**.
>
> ### Weakness 2. Figure 3b sample size
>
> Thanks for raising this question. We hypothesize that larger networks to generate more samples in the last layer may lead to the overfitting issue or increase the training instability. A similar insight can refer to [1] that points out deeper RL networks fail, arising from the training instability. Similar issues also exist in QR-DQN and MMDDRL algorithms, i.ei., the number of quantiles and particles.
>
> ### Reproducibility. 3 Seeds.
> Almost all previous distributional RL algorithms, including C51, QRDQN, and MMDDRL, conducted their Atari experiments over 3 seeds as it has been very costly in computation across all 55 Atari games. We sincerely hope you can understand this fact. To the best of our knowledge, combined with the significant confidence intervals shown in Figure 1, three seeds ought to be sufficient to make a comparison among algorithms, although we agree with you that more seeds can be better.
>
> [1] Nils Bjorck, Carla P. Gomes, and Kilian Q. Weinberger. Towards deeper deep reinforcement learning with spectral normalization. (NeurIPS 2021)

---

> > ### Comment · Reviewer_vqCE · 2022-11-10
> > **Reviewer's Response to Author Rebuttal**
> >
> > Thank you for addressing my comments.
> >
> > I agree that this paper has its own merits since it uses a new metric of Sinkhorn divergence to compare the distributions in  the DRL framework. I believe that this method has to be documented for sure, even if the proposed method does not significantly outperform the existing methods.
> >
> > Since ICLR has its own quirks/standards and based on my own experience of submitting and reviewing papers, I don't think this work qualifies for this conference (or the other way around). However, I won't mind if this work is accepted in this conference.
> >
> > Regarding the reproducibility, I only asked to present a few figures in main text with more seeds. It does not have to be for all the 55 environments since you can't include all 55 in the main text!

---

> > > ### Comment · Area_Chair_VpBE · 2022-11-20
> > > **What is the main reason for not qualifying for ICLR?**
> > >
> > > Dear Reviewer vqCE,
> > >
> > > Thank you very much for the discussion.  I understand your evaluation, but what is the main reason that you don't think this work qualifies for this conference?  Is it because "the superiority over existing state-of-the-art algorithms may not be sufficiently significant"?

---

> > > > ### Comment · Reviewer_vqCE · 2022-11-21
> > > > **Main reason for not qualifying for ICLR**
> > > >
> > > > Yes,  "the superiority over existing state-of-the-art algorithms may not be sufficiently significant" is the reason according to me.
> > > > Thanks.

---

### Author Response · Authors · 2022-11-08
**General Response**

Dear Reviewers and ACs,

Firstly, we are grateful for all the valuable comments and suggestions we received. We have responded to all reviewers respectively and revised the paper accordingly. We also add the **original paper along with the source code in the supplementary file for reference**. Please be free to let us know if there is any further question and we are happy to respond further.

Best Regards,
Authors

---

### Decision · Program_Chairs · 2023-01-20

**Decision:**

Reject

**Justification For Why Not Higher Score:**

The proposed approach does not significantly outperform existing methods, and it is unclear when and why one wants to use the proposed approach.

**Justification For Why Not Lower Score:**

N/A

**Metareview: Summary, Strengths And Weaknesses:**

This paper proposes a new distributional reinforcement learning (DRL) method, which uses Sinkhorn divergence as a measure of the divergence between distributions, and thus interpolates Wasserstein distance and Maximum Mean Discrepancy, which are commonly used in standard DRL today.

The paper establishes the contraction property of the Sinkhorn iteration, which is an essential property for DRL and thus constitutes the main strength of the paper.

However, the empirical advantage of the new DRL with Sinkhorn divergence over existing DRL methods are limited, and it is unclear when and why one might want to use the proposed approach over existing methods.